# MOSIV: Multi-Object System Identification from Videos

Chunjiang Liu[λ]  Xiaoyuan Wang[λ*] Qingran Lin[δ*] Albert Xiao[λ]  Haoyu Chen[σ]  Shizheng Wen[π]
Hao Zhang[μ]  Lu Qi[ν]  Ming-Hsuan Yang[τ]  Laszlo A. Jeni[λ†] Min Xu[λ†] Yizhou Zhao[λ†]

[λ] CMU  [δ] Georgia Tech  [σ] Harvard  [π] ETH Zurich  [μ] UIUC  [ν] Insta360  [τ] UC Merced

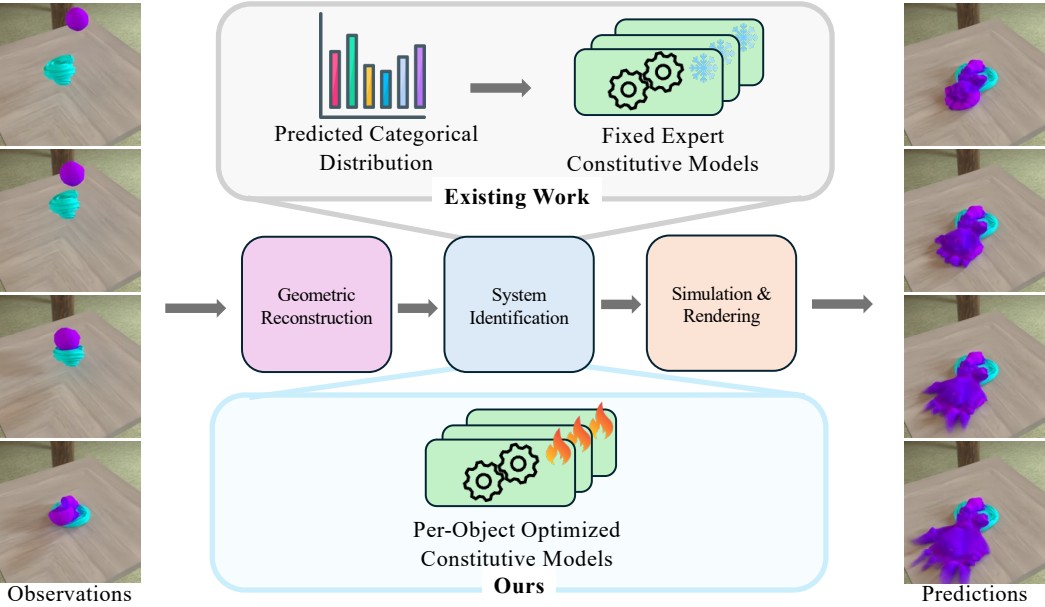

Figure 1: From multi-view observations of multi-object scenes (left), prior approaches select from a fixed library of expert constitutive models via categorical prediction, leading to visually implausible and weakly calibrated physics dynamics. **MOSIV** instead performs geometric reconstruction, per-object system identification of continuous constitutive parameters, enabling both faithful reproduction of observed interactions and accurate prediction of future behaviors (right).

## Abstract

We introduce the challenging problem of multi-object system identification from videos, for which prior methods are ill-suited due to their focus on single-object scenes or discrete material classification with a fixed set of material prototypes. To address this, we propose MOSIV, a new framework that directly optimizes for continuous, per-object material parameters using a differentiable simulator guided by geometric objectives derived from video. We also present a new synthetic benchmark with contact-rich, multi-object interactions to facilitate evaluation. On this benchmark, MOSIV substantially improves grounding accuracy and long-horizon simulation fidelity over adapted baselines, establishing it as a strong baseline for this new task. Our analysis shows that object-level fine-grained supervision and geometry-aligned objectives are critical for stable optimization in these complex, multi-object settings. The source code and dataset will be released.

---

*Equal contribution. †Equal advisorship.

# 1 INTRODUCTION

Real-world scenes are dynamic and often chaotic; multiple objects collide, slide, and reconfigure themselves through a constant dance of contact. Most methods (Cai et al., 2024; Zhao et al., 2025; Liang et al., 2019; Raissi et al., 2019; Li et al., 2023a; 2022) that try to understand an object's physics from video are designed for simple, controlled settings—typically a single object moving in isolation. These approaches fail in complex, everyday environments where objects bump into one another, block each other from view, and have their motions intricately linked. To enable advanced applications like robotic manipulation in cluttered spaces (Yin et al., 2021; Shi et al., 2023; 2024a) or physically plausible scene editing, we need a method that can learn the physical properties of all objects and their interactions simultaneously, just by watching videos of them.

We introduce and formalize this challenge as *multi-object system identification from videos*. Our goal is straightforward: given multi-view videos of interacting objects, we aim to reconstruct their changing 4D geometry (3D shape over time) and identify the physical properties of each object—such as its stiffness, plasticity, and friction. A successful result is a "digital twin" of the scene, where a physics simulator can reproduce the observed motion, accurately predict future interactions, and generalize to novel scenarios, such as different initial conditions or force fields.

Object interactions are a double-edged sword: they provide rich signals that make hidden physical properties observable, but they also create challenges like occlusions and abrupt, complex motions. Solving this multi-object problem requires an accurate 4D reconstruction (Kratimenos et al., 2024), a simulator that can handle contact and friction between different materials (De Vaucorbeil et al., 2020), and a learning process focused on identifying specific parameters rather than just selecting a material category. Ambiguities like distinguishing stiffness from friction can't be resolved by appearance alone; the system must analyze geometry and motion over time (Cai et al., 2024).

We compare our method to OMNIPHYSGS (Lin et al., 2025), a baseline chosen for its ability to model scenes with varying materials. However, its core design is ill-suited for our task. OM-NIPHYSGS performs model selection—it classifies materials by picking from a small, fixed library—rather than identifying the continuous parameters (e.g., $E, \nu, \mu$) needed for accurate physics. In contrast, our method learns continuous parameter maps for each object and couples this representation with a differentiable MPM simulator (Jiang et al., 2016; Hu et al., 2018; Geilinger et al., 2020; Du et al., 2021; Qiao et al., 2021) that accurately models how different materials interact, yielding identifiable parameters and realistic dynamics. We also compare COUPNERF (Li et al., 2024a), which tackles multi-object system identification with an implicit NeRF representation under a free-fall regime. While effective in that setting, time-optimized NeRF fields are computationally heavy and prone to temporal inconsistency in contact-rich, highly deformable scenes.

Our solution is built on three synergistic components. First, object-aware dynamic Gaussians track each object's unique material properties in 4D with pre-defined 2D material masks. Second, a differentiable Material Point Method (MPM) simulator accurately models complex inter-material physics, including contact and friction. Third, joint multi-object fitting learns continuous parameters by aligning simulated surfaces and silhouettes with visual evidence from the video. To validate this approach, we formalize the task, release a new synthetic dataset, and adapt the OMNIPHYSGS and COUPNERF baseline to use direct visual supervision for a fair comparison.

On contact-rich scenes, our method significantly reduces parameter error and improves simulation accuracy over time compared to the adapted baseline. Our simulated trajectories remain stable and aligned with the observed video, whereas the baseline's discrete material choices cause drift. Ablation studies confirm that all three components are essential for achieving stability and accuracy.

To sum up, the main contributions of this work are threefold:

- We formalize the task of multi-object system identification from videos and release a challenging synthetic dataset with ground-truth physical parameters to drive future research.
- We propose a new framework that combines object-aware dynamic Gaussians with joint multi-object fitting. This approach uses geometry-driven supervision to directly identify the continuous, object-specific physical properties from video.
- We validate our method on the new dataset, demonstrating state-of-the-art performance. Our approach surpasses the OMNIPHYSGS-based baseline in identifying material parameters and achieves significantly higher physical accuracy and visual fidelity in simulations.

## 2 RELATED WORK

**Dynamic Reconstruction.** Dynamic 4D reconstruction aims to recover temporally varying, high-fidelity geometry and appearance from single or multi-view video. Implicit models, such as Neural Radiance Fields (NeRF) (Mildenhall et al., 2021), have been a foundational approach for novel-view synthesis. To handle motion, techniques extend NeRF with explicit deformation fields (Pumarola et al., 2021) or regularize them with priors on volume and topology (Park et al., 2021a;b). While these implicit models excel at novel-view synthesis, they often yield noisy or poorly conditioned geometry, limiting downstream physical analysis (Li et al., 2023b). Later, the advent of 3D Gaussian Splatting (3DGS) (Kerbl et al., 2023) introduced a fast, explicit representation and catalyzed a wave of dynamic methods that either reconstruct each frame independently (Luiten et al., 2024; Wu et al., 2024a) or learn a canonical set of Gaussians that deform over time (Yang et al., 2024; Kratimenos et al., 2024; Wang et al., 2025). These approaches improve real-time rendering and geometric stability compared to purely implicit fields, but they typically do not encode physical laws.

**Dynamic Simulation.** The intersection of perception and physics has led to methods that infuse physical structure into generative pipelines. Approaches often couple text and video generative models with 3D representations to synthesize dynamic scenes (Bahmani et al., 2024; Ling et al., 2024; Ren et al., 2023; Singer et al., 2023). Other work treats Gaussian kernels as both visual and physical primitives, embedding Newtonian dynamics or constitutive behavior to enable constrained rendering and simulation (Xie et al., 2024; Liu et al., 2024a; Lin et al., 2025; Li et al., 2023b; Qiu et al., 2024; Borycki et al., 2024; Zhong et al., 2024; Fu et al., 2024). For instance, Gaussian Splashing (Feng et al., 2025) integrates position-based dynamics within 3DGS to handle solids, fluids, and deformables. A complementary trend—motion-conditioned simulation—steers object trajectories using learned priors rather than explicit solvers (Li et al., 2024b; Geng et al., 2025; Wang et al., 2024; Wu et al., 2024b; Shi et al., 2024b). More recent "neural physics" approaches infer dynamics from video or generative priors to synthesize physically plausible motion without dedicated simulators (Zhang et al., 2024; Liu et al., 2024b; Huang et al., 2025; Feng et al., 2024; Tan et al., 2024). Despite promising results, many methods specialize to limited material families or rely on priors trained for visual fidelity rather than faithful mechanics, which hinders generalization.

**System Identification From Videos.** System identification in vision and robotics seeks to infer latent physical laws and material properties directly from visual observations (Li et al., 2023a; Liang et al., 2019; Raissi et al., 2019; Sundaresan et al., 2022; Li et al., 2022; Zhong et al., 2024; Zheng et al., 2024b), a capability that underpins realistic simulation and effective robot interaction with deformable and elasto-plastic objects (Shi et al., 2023; 2024a; Liang et al., 2024; Zheng et al., 2024a; Qiao et al., 2022). Classical approaches leverage explicit simulators such as FEM or mass–spring systems (Takahashi & Lin, 2019; Wang et al., 2015), but they typically assume known geometry and limited material families. Data-driven alternatives learn dynamics or parameters from experience (Sanchez-Gonzalez et al., 2020; Li et al., 2018; Xu et al., 2019), improving flexibility yet often struggling to generalize across unseen materials and conditions. Differentiable physics has narrowed this gap by enabling end-to-end gradient-based estimation through simulators (Hu et al., 2019; Huang et al., 2021; Chen et al., 2022; Du et al., 2021; Geilinger et al., 2020; Heiden et al., 2021; Jatavallabhula et al., 2021; Ma et al., 2022; Qiao et al., 2021; Kaneko, 2024), though accurate modeling and priors remain crucial. Recent directions fuse neural representations with physics, either learning neural constitutive laws that augment expert simulators (Ma et al., 2023; Cao et al., 2024) or adopting hybrid formulations (e.g., MPM/spring–mass with 3D Gaussian splatting) to reconstruct geometry and identify material properties from video (Li et al., 2023b; Cai et al., 2024; Zhong et al., 2024; Shao et al., 2024). CoupNeRF(Li et al., 2024a) uses a hybrid approach that combines an implicit NeRF representation with differentiable MPM to perform multi-object systemID.

## 3 METHOD

### 3.1 PROBLEM STATEMENT

We consider a scene with $K$ deformable objects observed as multi-view RGB videos over $T$ timestamps. Each object may contain one material class drawn from a material set $\mathcal{M}$. The goal is to recover (i) a simulation-ready continuum for all objects across all time and (ii) a collection of per-material parameters $\boldsymbol{\Theta} = \{\boldsymbol{\theta}_m\}_{m \in \mathcal{M}}$ such that a forward simulator reproduces the observed motion and predicts future motions over long horizons. The estimator uses only videos, camera calibra-

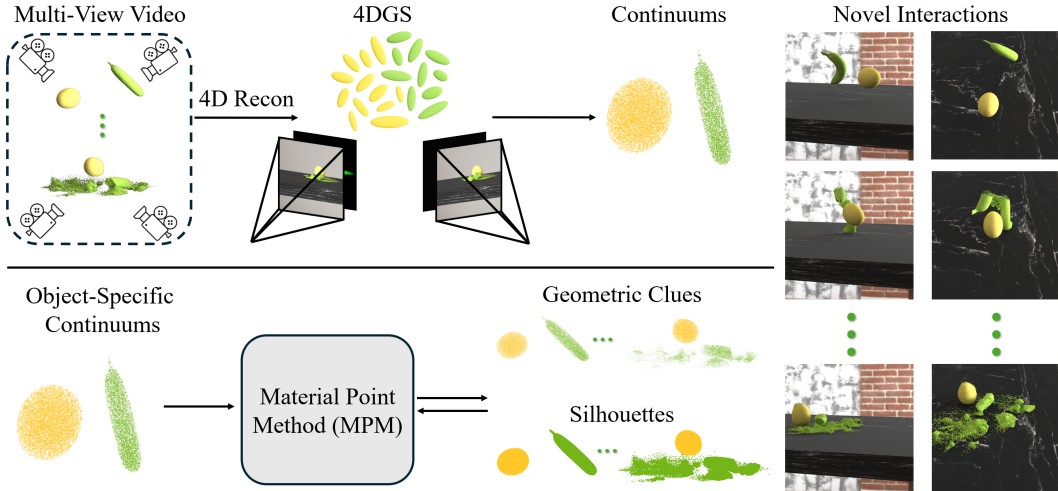

Figure 2: **(1) Geometric reconstruction.** From multi-view RGB videos, we reconstruct object geometry and disentangle material-specific motion via optimizing 4D Gaussian Splatting (4DGS) with object masks. **(2) Continuum simulation.** The reconstructed Gaussians are lifted into object-specific continuums, which serve as the initial states for a differentiable MPM. Geometry-aligned losses on surfaces and silhouettes drive physics parameter optimization under inter-material contact and friction. **(3) Applications.** The calibrated model generalizes to novel interaction scenarios, enabling physically faithful rollouts and long-horizon predictions of complex multi-object dynamics.

tion, and instance masks extracted from the videos. Unlike single-object settings, our formulation estimates parameters *independently for each object instance*. That is, every object in the scene is equipped with its own set of material parameters, which are optimized directly from its geometry and motion cues. This per-instance treatment avoids the need for pre-defined material sharing across objects, and enables our pipeline to handle multiple objects even under complex contact.

### 3.2 OVERVIEW

The pipeline in Fig. 2 proceeds in three stages. First, we reconstruct an object-aware dynamic Gaussian field from the multi-view video, with additional supervision from labeled 2D material masks that indicate the material type of each object. Second, following GIC (Cai et al., 2024), a compact Gaussian-to-continuum lifting converts each object's reconstruction into a simulation particle set, where particles carry positions, a material-family label, and shared material parameters. Third, starting from this particle state, the differentiable MPM is rolled out over the observed frames. Our geometry-aligned objectives then compare these simulated surfaces and silhouettes to those extracted from the reconstructed Gaussians (via per-object 3D Chamfer and 2D alpha-mask losses), and back-propagate through the MPM to jointly optimize the unknown physical parameters $\boldsymbol{\Theta}$.

#### 3.2.1 PRELIMINARIES: MATERIAL POINT METHOD

We evolve a set of $Q$ material points with a differentiable time-stepping map $\mathbf{z}_{n+1} = \mathcal{T}(\mathbf{z}_n; \boldsymbol{\Theta})$, $n = 0, \dots, N-1$, where $\mathbf{z}_n = \{\mathbf{x}_n(i), \mathbf{v}_n(i), \mathbf{F}_n^e(i)\}_{i=1}^{Q}$ contains position, velocity, and the elastic part of the deformation gradient at step $n$, and $N = T/\tau$ with simulation step size $\tau$ chosen so that $N \gg T$. Each step transfers particle states to a background grid, evaluates stresses via an elastic law $\mathcal{E}$ to obtain the first Piola–Kirchhoff stress $\mathbf{P}_n$, updates momenta and velocities on the grid, and transfers back to particles. A plastic projection $\mathcal{P}$ maps the trial elastic tensor to an admissible $\mathbf{F}_{n+1}^e$. Grid-level contact and Coulomb friction resolve interactions between objects and materials. The resulting map $\mathcal{T}$ is fully differentiable with respect to both state and parameters, thereby enabling efficient gradient-based identification and optimization.

#### 3.2.2 PRELIMINARIES: DYNAMIC GAUSSIAN RECONSTRUCTION

We represent the scene with canonical Gaussian kernels that are warped in time by a low-rank deformation. Let $\mathcal{G}_0 = \{(\boldsymbol{\mu}, r, \mathbf{c}, \sigma)\}$ denote kernel centers, isotropic scales, colors, and opacities.

A pair of networks produces temporal bases and spatially varying gates, yielding for each time $t$

$$\boldsymbol{\mu}_t = \boldsymbol{\mu} + \sum_{b=1}^{B} \alpha_b(\boldsymbol{\mu})\,\boldsymbol{\psi}_b^{\mu}(t), \qquad r_t = r + \sum_{b=1}^{B} \alpha_b(\boldsymbol{\mu})\,\psi_b^{r}(t), \tag{1}$$

with $\boldsymbol{\psi}_b^{\mu} \in \mathbb{R}^3$ and $\psi_b^r \in \mathbb{R}$. We optimize photometric agreement across views,

$$\min_{\mathcal{G}_0,\text{net}} \mathcal{L}_1(\hat{\mathbf{I}}_t, \mathbf{I}_t) + \lambda_{\text{SSIM}}\mathcal{L}_{\text{SSIM}}(\hat{\mathbf{I}}_t, \mathbf{I}_t) + \lambda_r \|r_t\|_1, \tag{2}$$

where $\hat{\mathbf{I}}_t$ are rendered frames. Instance masks partition kernels by object; material masks, when available or synthesized, partition kernels by material. These labels are passed to the simulator.

### 3.3 Gaussian-to-continuum lifting in the multi-object regime

Dynamic Gaussians are optimized for rendering and are spatially nonuniform. We therefore derive simulation particles from a thin occupancy field per object. For each object $k$, We generate a rough internal shape by randomly sampling particles within the bounding box of Gaussian points and retaining only those that align with the object's depth as rendered from multiple camera views. Then we construct a density field that progressively increases in resolution. In each iteration, we upsample the grid, smooths the field (mean filtering) to blur boundaries, and reassign high density to voxels containing actual particles to prevent the smoothing process from eroding the object's true shape. Finally, the specific object surface is isolated by applying a threshold to this high-resolution, refined density field.

Compared to single-object lifting, the multi-object setting requires two additional constraints. First, we enforce disjoint supports between objects at initialization by assigning overlapping voxels to the nearest object surface and removing residual interpenetrations. Second, we maintain material labels on particles and ensure that per-object grids use a compatible resolution so that interfaces align at contact. The output is a set of particles $\tilde{\mathcal{P}}_k(0)$ with per-particle object and material tags; we use these particles only for shape rendering and as the initial state for simulation.

### 3.4 Multi-material parameterization and contact

Each material $m \in \mathcal{M}$ is associated with a parameter vector $\boldsymbol{\theta}_m$ controlling its elastic, plastic, and viscous response. To reduce degrees of freedom while capturing inter-material behavior, we model Coulomb friction as an interface between materials $m$ and $m'$ using a symmetric composition $\mu_{m,m'} = g(\mu_m, \mu_{m'})$ with $g(a,b) = \frac{1}{2}(a+b)$, although a fully pairwise parameterization is also supported when annotations allow. We assign parameters on a per-object basis: each object instance $k$ carries its own parameter vector $\boldsymbol{\theta}_k$, which governs its elastic, plastic, and frictional response. Even if two objects correspond to the same real-world material, we do not impose parameter sharing; identifiability emerges from object-wise geometry and silhouette constraints under interaction. This per-instance treatment ensures flexibility when objects deform or respond differently under contact.

### 3.5 Geometry-aligned objectives for multi-object identification

For each camera $j$ at t observed times $\{t_i\}_{i=1}^m$, we render silhouettes $A_{j,k}(t_i)$ per object $k$ and compare to target silhouettes $\tilde{A}_{j,k}(t_i)$. We also compare simulated and extracted surfaces $S_k(t_i)$ and $\tilde{S}_k(t_i)$ via a symmetric Chamfer distance. The overall objective is

$$\mathcal{L}_{\text{ID}} = \frac{1}{m}\sum_{i=1}^m \left[ \sum_{k=1}^K \mathcal{L}_{\text{CD}}\big(S_k(t_i), \tilde{S}_k(t_i)\big) + \frac{1}{n}\sum_{j=1}^n \sum_{k=1}^K \mathcal{L}_1\big(A_{j,k}(t_i), \tilde{A}_{j,k}(t_i)\big) \right]. \tag{3}$$

### 3.6 Optimization in the multi-object setting

Training occurs in three stages. Stage I reconstructs dynamic Gaussians and assigns instance partitions using object masks. Stage II converts each object's reconstruction into a simulation-ready continuum via Gaussian-to-continuum lifting. Stage III optimizes the per-object parameter vectors $\{\boldsymbol{\theta}_k\}_{k=1}^K$ by minimizing equation 3 through MPM. To stabilize training, we adopt a horizon curriculum that gradually increases the rollout length as alignment improves, and use an alternating update strategy that interleaves parameter optimization with occasional re-synchronization of particle state to reduce drift. All experiments follow this fixed schedule unless otherwise noted.

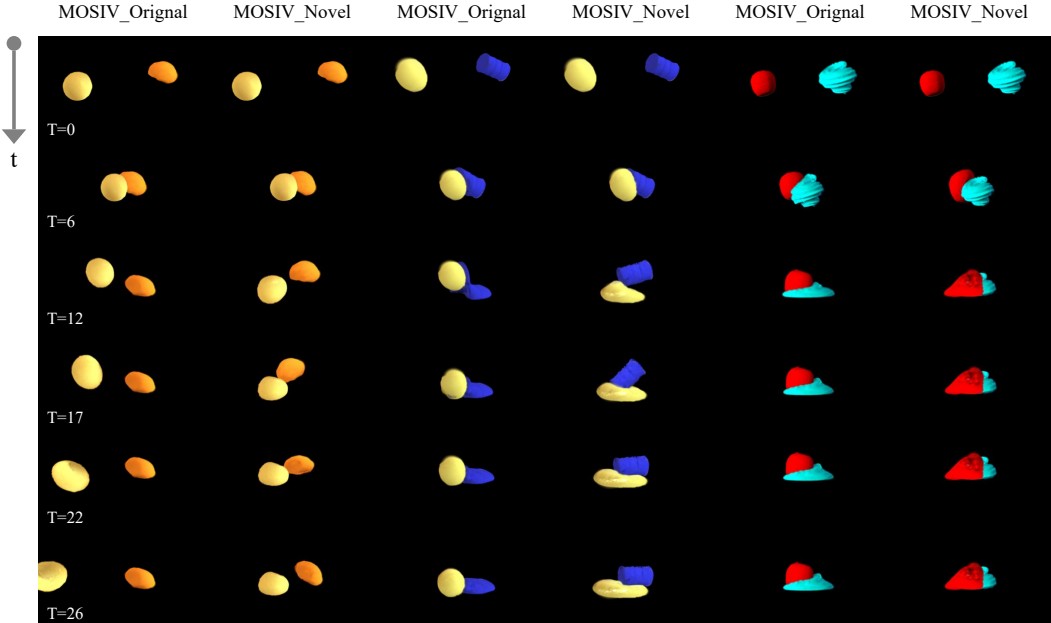

Figure 3: **Novel Interaction.** Left—MOSIV original GT video sequence. Right—rollout after swapping object physics parameters while keeping initial conditions unchanged. Rows show time.

### 3.7 Novel Interactions

We enable multi-object novel interactions by varying initial conditions and material assignments. Novel interactions can arise from changes in velocities, object placements, or physical properties. Since existing datasets already span diverse velocity and impact settings, we focus on the material dimension. Specifically, for each sequence we keep geometry, poses, and velocities fixed, and permute the identified per-object constitutive parameters to create new material assignments. We then roll out the differentiable MPM to predict the resulting dynamics. As shown in Fig. 3, these material swaps yield distinct yet physically plausible outcomes consistent with the reassigned stiffness, yield, and friction, demonstrating MOSIV's capacity to predict behaviors beyond observed interactions.

## 4 Experiments

### 4.1 Experimental Setting

**Datasets and evaluation protocol** To evaulate our system identification methods, we generate a new multi-object dataset using the Genesis physics platform (Xian et al., 2024), an engine that supports simulation via the Material Point Method. This dataset is composed of 45 multi-view videos of two-object interactions. Specifically, we use 10 unique geometries (egg, pawn, apple, bread, cream, barrel, potato, cushion, banana, mushroom), and 5 materials (elastic, elastoplastic, liquid, sand, snow), and we assign each material to two objects. For each pair of objects, we intialize them to a random position and rotation, then set the initial velocities such that the two objects collide at a specified time. The objects are then set in motion, collide, free-fall due to gravity, and eventually land on a flat table surface. We capture 30 frames of this interaction with 11 camera views evenly spaced around the hemisphere above the table. For enhanced photorealism, we use 10 different background environments, 12 different table textures, and a realistic color for each object. We provide example views from some selected sequences in the Appendix.

**Baselines** We adapt **OmniPhysGS-RGB** to the video-driven SysID setting as follows: (1) initialize the reconstruction using the fused input point cloud at the first frame; (2) keep the decoder and expert-library architecture unchanged; (3) replace the original SDS objective with an image-space photometric loss as equation 1. We further construct an oracle variant of OmniPhysGS-RGB, named with **OmniPhysGS-RGB w/ Oracle**, to isolate the impact of material model selection. Instead of requiring OmniPhysGS to infer the correct constitutive models, we directly provide it with the ground-truth per-object models while keeping the rest of the architecture and training setup unchanged. This gives an upper-bound reference by removing the challenge of material identification.

| | Method | Inter-Material Interaction | | | | | | Intra-Material Interaction | | | | Average |
|---|---|---|---|---|---|---|---|---|---|---|---|---|
| | | E–P | E–F | E–S | P–F | P–S | F–S | E–E | P–P | F–F | S–S | |
| PSNR ← | OPGS (Lin et al., 2025) | 27.63 | 27.01 | 24.46 | 26.24 | 26.80 | 24.89 | 26.84 | 29.79 | 23.59 | 24.72 | 25.93 |
| | OPGS w/ Oracle (Lin et al., 2025) | 25.37 | 24.62 | 23.26 | 23.81 | 25.98 | 23.36 | 25.06 | 25.86 | 22.53 | 24.52 | 24.39 |
| | MOSIV (Ours) | **30.89** | **30.29** | **26.57** | **32.21** | **29.07** | **29.88** | **27.96** | **36.16** | **35.16** | **26.87** | **30.51** |
| SSIM ← | OPGS (Lin et al., 2025) | 0.968 | 0.966 | 0.892 | 0.971 | 0.945 | 0.951 | 0.951 | 0.980 | 0.953 | 0.931 | 0.945 |
| | OPGS w/ Oracle (Lin et al., 2025) | 0.952 | 0.941 | 0.877 | 0.949 | 0.938 | 0.933 | 0.948 | 0.955 | 0.936 | 0.931 | 0.930 |
| | MOSIV (Ours) | **0.983** | **0.982** | **0.945** | **0.986** | **0.973** | **0.977** | **0.970** | **0.992** | **0.987** | **0.971** | **0.977** |
| CD → | OPGS (Lin et al., 2025) | 11.10 | 3.931 | 27.97 | 2.692 | 10.16 | 8.165 | 23.82 | 1.030 | 7.281 | 13.85 | 11.79 |
| | OPGS w/ Oracle (Lin et al., 2025) | 33.24 | 81.98 | 46.09 | 91.28 | 17.33 | 43.18 | 10.35 | 77.54 | 43.82 | 3.01 | 43.50 |
| | MOSIV (Ours) | **1.095** | **0.358** | **2.022** | **0.183** | **0.839** | **0.593** | **4.876** | **0.129** | **0.166** | **2.301** | **1.256** |
| EMD → | OPGS (Lin et al., 2025) | 0.085 | 0.078 | 0.135 | 0.069 | 0.085 | 0.092 | 0.134 | 0.052 | 0.105 | 0.104 | 0.095 |
| | OPGS w/ Oracle (Lin et al., 2025) | 0.157 | 0.227 | 0.188 | 0.243 | 0.112 | 0.186 | 0.113 | 0.228 | 0.199 | **0.063** | 0.168 |
| | MOSIV (Ours) | **0.043** | **0.041** | **0.069** | **0.028** | **0.047** | **0.052** | **0.064** | **0.012** | **0.033** | 0.103 | **0.049** |

Table 1: **Observable state simulation on MOSIV Synthetic dataset.** Columns are grouped by material-pair types. Material abbreviations: E (elastic), P (plastic), F (fluid), S (sand).

**Metrics**  We report standard geometric and photometric measurements. Discrepancies between reconstructed and ground-truth point sets are measured by *Chamfer Distance (CD)* (Ma et al., 2020) (reported in $10^3$ mm$^2$) and *Earth Mover's Distance (EMD)*. On sequences with reference renderings (e.g., Spring-Gaus), we assess frame fidelity using *PSNR* (Hore & Ziou, 2010) and *SSIM* (Wang et al., 2004) to quantify how well predicted states match future observations.

**Implementation details**  Our dynamic Gaussian module follows the design in (Kratimenos et al., 2024) (motion backbone: 8 fully connected layers; 10 lightweight heads output per-basis residuals for centers and scales; coefficient network: 4 fully connected layers). Training uses the photometric objective in Eq. equation 2. We employ compact object-wise occupancy refinement to obtain simulation particles; overlapping voxels at initialization are assigned to the nearest object surface to ensure disjoint supports, and per-object grids are aligned in resolution so that contact interfaces match. We adopt a differentiable MPM simulator with a time step $\tau = 1/4800$ (200 substeps per 24 fps frame) and a grid resolution of $4096^3$. Model parameters are optimized with the Adam optimizer, performing 80 iterations for initial velocity estimation followed by 200 iterations for physical property refinement. All experiments are conducted on an NVIDIA RTX A6000 GPU (48 GB).

### 4.2 Quantitative Results Comparison

**Observable state simulation.**  In Tab. 1, we present a quantitative comparison between MOSIV, OmniPhysGS-RGB w/ Oracle, and OmniPhysGS-RGB on the task of observable state simulation. As shown in the table, MOSIV consistently and substantially outperforms both OmniPhysGS-RGB and OmniPhysGS-RGB w/ Oracle across all reported metrics. This highlights MOSIV 's ability to accurately reconstruct and predict object dynamics even in scenes that feature a wide variety of material properties and complex physical interactions. These results underscore the robustness and capacity of MOSIV for challenging real-world dynamics.

**Future state simulation.**  In Tab. 2, we evaluate MOSIV on challenging tasks of future state simulation, where the objective is to forecast long-term scene evolution beyond the observed frames. MOSIV clearly surpasses OmniPhysGS-RGB and OmniPhysGS-RGB w/ Oracle across all reported metrics, highlighting its capability to anticipate object trajectories under complex physical interactions and diverse material compositions. This improvement reflects the method's accurate system identification, effectively inferring each object's geometry, dynamic behavior, and underlying physical properties. Such understanding of both scene structure and physics enables MOSIV to generalize well, delivering reliable predictions in previously unseen and physically intricate scenarios.

### 4.3 Qualitative Results Comparison

**Observable and Future state simulation.**  Fig. 4 compares Ground Truth, MOSIV, OmniPhysGS-RGB w/ Oracle, and OmniPhysGS-RGB on two representative scenes—plastcine–fluid (P–F) and sand–sand (S–S). Across the observed frames, MOSIV better preserves object geometry and contact boundaries: fluids do not over-spread, sand clusters remain compact, and plastic bodies retain plausible deformation. Baselines show blur, shape erosion, and contact leakage. In the predicted

| | Method | Inter-Material Interaction | | | | | | Intra-Material Interaction | | | | Average |
|---|---|---|---|---|---|---|---|---|---|---|---|---|
| | | E–P | E–F | E–S | P–F | P–S | F–S | E–E | P–P | F–F | S–S | |
| PSNR ← | OPGS (Lin et al., 2025) | 20.65 | 20.68 | 16.40 | 20.26 | 19.97 | 18.03 | 18.40 | 25.19 | 21.48 | 16.31 | 19.00 |
| | OPGS w/ Oracle (Lin et al., 2025) | 19.58 | 18.91 | 15.82 | 17.83 | 19.12 | 16.43 | 17.89 | 21.14 | 18.91 | 18.37 | 17.97 |
| | MOSIV (Ours) | **25.57** | **27.58** | **22.83** | **29.27** | **28.34** | **28.92** | **22.79** | **37.20** | **35.47** | **24.63** | **28.26** |
| SSIM ← | OPGS (Lin et al., 2025) | 0.947 | 0.941 | 0.741 | 0.954 | 0.895 | 0.914 | 0.902 | 0.970 | 0.942 | 0.847 | 0.888 |
| | OPGS w/ Oracle (Lin et al., 2025) | 0.928 | 0.916 | 0.721 | 0.927 | 0.884 | 0.884 | 0.907 | 0.939 | 0.895 | 0.851 | 0.869 |
| | MOSIV (Ours) | **0.970** | **0.975** | **0.891** | **0.980** | **0.966** | **0.971** | **0.942** | **0.994** | **0.984** | **0.956** | **0.963** |
| CD → | OPGS (Lin et al., 2025) | 108.1 | 15.54 | 131.5 | 5.620 | 27.53 | 18.80 | 149.9 | 2.181 | 11.28 | 43.45 | 51.92 |
| | OmniPhysGS w/Oracle (Lin et al., 2025) | 151.76 | 455.23 | 235.67 | 448.43 | 53.49 | 200.27 | 38.33 | 461.80 | 279.00 | 12.09 | 215.83 |
| | MOSIV (Ours) | **5.939** | **0.532** | **9.31** | **0.255** | **1.151** | **1.141** | **16.59** | **0.132** | **0.183** | **1.867** | **3.710** |
| EMD → | OPGS (Lin et al., 2025) | 0.258 | 0.161 | 0.346 | 0.101 | 0.144 | 0.155 | 0.396 | 0.082 | 0.123 | 0.207 | 0.199 |
| | OPGS w/ Oracle (Lin et al., 2025) | 0.361 | 0.600 | 0.497 | 0.600 | 0.211 | 0.456 | 0.260 | 0.600 | 0.500 | 0.123 | 0.408 |
| | MOSIV (Ours) | **0.081** | **0.048** | **0.135** | **0.029** | **0.062** | **0.061** | **0.140** | **0.019** | **0.035** | **0.102** | **0.071** |

Table 2: **Future state simulation on MOSIV Synthetic dataset.** Columns are grouped by material-pair types. Material abbreviations: E (elastic), P (plastic), F (fluid), S (sand).

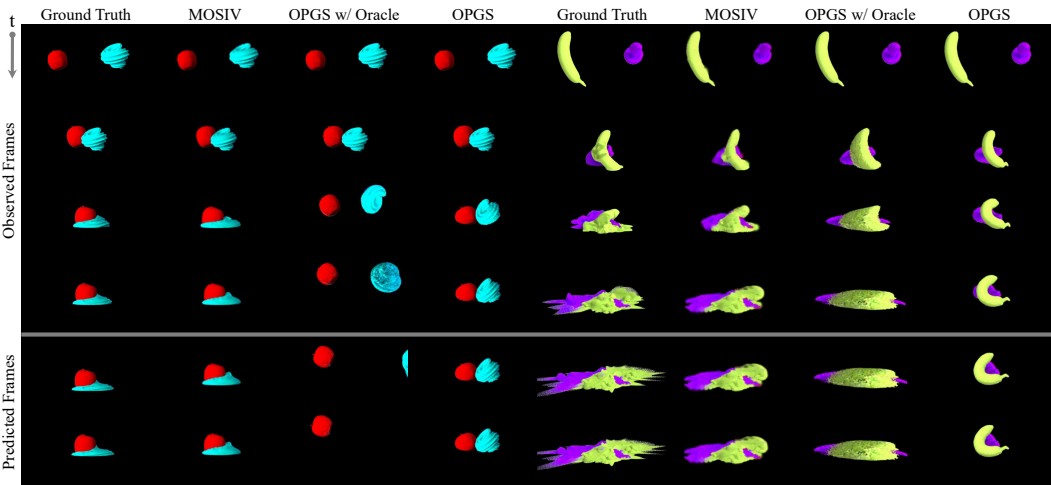

Figure 4: **Qualitative comparison of multi-object interactions.** The first four columns shows a **plasticine–fluid (P–F)** example; the last four columns shows a **sand–sand (S–S)** example.

frames of Fig. 4, MOSIV sustains stable long-horizon rollouts: collision timing and post-impact trajectories remain consistent with Ground Truth. OmniPhysGS-RGB variants drift over time: Fluids overshoot and sands disperse unrealistically, indicating weaker identification of the system and contact handling. Fig. 5 compares Ground Truth, CoupNeRF* and MOSIV on two highly dynamic scenes with significant deformations—plastcine–sand (P–S) and elastic–plastic (E–P). In the P–S scene, CoupNeRF* does not produce the correct physics: both plasticine and sand behave like a viscous fluid, losing the expected distinction between granular flow for sand and plastic deformation for plasticine. In the E–P scene, CoupNeRF* also deviates from the correct behavior and the appearance is also distorted. By contrast, MOSIV preserves the expected material-specific dynamics and maintains consistent appearance across frames.

**Trajectory comparison.** Fig. 6 visualizes particle trajectories. MOSIV 's streamlines align tightly with Ground Truth, showing coherent paths through contact events and minimal accumulation error. In contrast, OmniPhysGS-RGB and OmniPhysGS-RGB w/ Oracle produce fragmented or biased paths and increasing drift. Overall, the qualitative results mirror the quantitative trends: MOSIV more faithfully captures contact-rich, multi-material dynamics over long horizons.

### 4.4 ABLATION: OBJECT-AWARE SUPERVISION VS. SCENE-WISE SUPERVISION

A central challenge in the multi-object regime is *association ambiguity* at contact: nearest-neighbor geometry and silhouette losses computed on the union of objects can spuriously *explain* a simulated

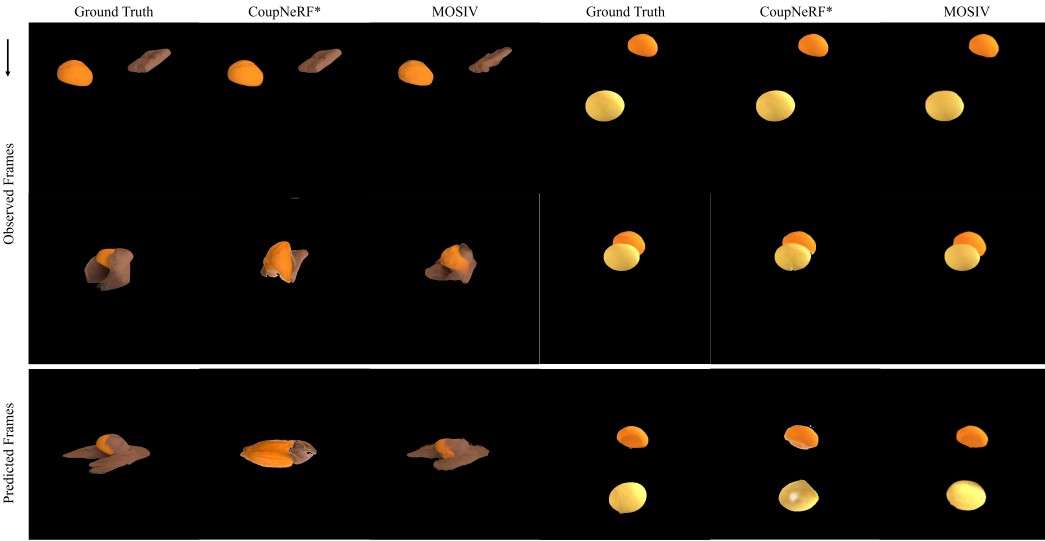

Figure 5: **Qualitative comparison between MOSIV and baselines.**

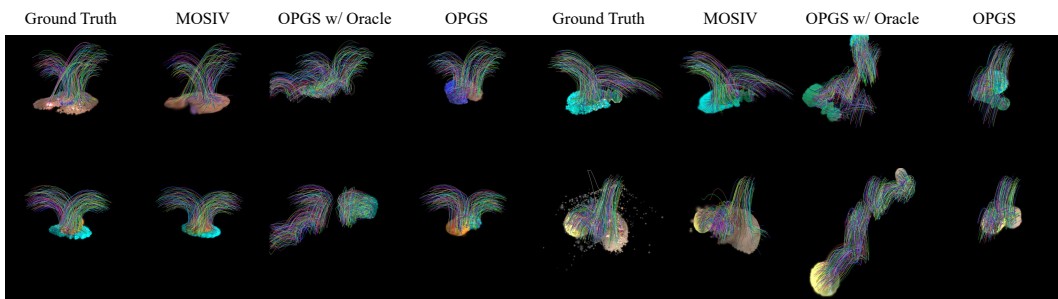

Figure 6: **Qualitative comparison of trajectory.** The trajectories illustrate how well each method captures long-term dynamics.

point on object $k$ by a ground-truth point on object $k'$ when the two bodies touch or interpenetrate in projection. This cross-object borrowing hides parameter miscalibration (e.g., an overly soft $k$ deforming into $k'$) and produces optimistic rollouts.

**Scene-wise losses (naive).** Let $\mathcal{P}^{\text{sim}}(t)$ and $\mathcal{P}^{\text{gt}}(t)$ denote the union of simulated and ground-truth surface samples at time $t$. The global Chamfer loss, $\mathcal{L}_{\text{CD}}^{\text{global}}(t) = d\big(\mathcal{P}^{\text{sim}}(t), \mathcal{P}^{\text{gt}}(t)\big) + d\big(\mathcal{P}^{\text{gt}}(t), \mathcal{P}^{\text{sim}}(t)\big)$, with $d(\cdot, \cdot)$ the one-sided nearest-neighbor distance, admits cross-object matches at contact. Similarly, a single alpha-mask loss per view, $\mathcal{L}_{\alpha}^{\text{global}}(t, j) = \big\|A_j^{\text{sim}}(t) - \tilde{A}_j(t)\big\|_1$, is blind to which object explains which pixels.

**Object-wise losses (ours).** We enforce supervision at the object level to preserve identities through contact. Let $\mathcal{P}_k^{\text{sim}}(t)$ and $\mathcal{P}_k^{\text{gt}}(t)$ be the simulated and target samples for object $k$, and $A_{j,k}^{\text{sim}}(t)$, $\tilde{A}_{j,k}(t)$ the per-object silhouettes at view $j$. Our geometry loss sums *disjoint* Chamfer distances: $\mathcal{L}_{\text{CD}}^{\text{obj}}(t) = \sum_{k=1}^{K}\Big[d\big(\mathcal{P}_k^{\text{sim}}(t), \mathcal{P}_k^{\text{gt}}(t)\big) + d\big(\mathcal{P}_k^{\text{gt}}(t), \mathcal{P}_k^{\text{sim}}(t)\big)\Big]$, and the silhouette loss aligns each object separately: $\mathcal{L}_{\alpha}^{\text{obj}}(t, j) = \sum_{k=1}^{K}\big\|A_{j,k}^{\text{sim}}(t) - \tilde{A}_{j,k}(t)\big\|_1$. This prevents the optimizer from trading deformation on one object against another to satisfy a global loss, yielding sharper gradients for contact mechanics and materially correct parameter updates. In practice, we find that object-aware supervision is crucial during impact and stick–slip transitions, where scene-wise losses can be minimized by swapping mass or stiffness across bodies in projection.

**Ablation Results.** We conduct the experiment on a subset of MOSIV, selecting six scenes in total, one from each inter-material interaction type. Table 3 shows that replacing scene-wise supervision

| Supervision Granularity | $\mathcal{L}_{\text{CD}}$ | $\mathcal{L}_\alpha$ | PSNR ↑ | SSIM ↑ | CD ↓ | EMD ↓ |
|---|---|---|---|---|---|---|
| | ✗ | ✓ | 26.59 | 0.964 | 53.21 | 0.132 |
| Scene-wise losses (naive) | ✓ | ✗ | 27.59 | 0.959 | 40.29 | 0.119 |
| | ✓ | ✓ | **27.89** | **0.968** | **22.13** | **0.091** |
| | ✗ | ✓ | 30.18 | 0.975 | 0.985 | 0.045 |
| Object-aware losses (ours) | ✓ | ✗ | 29.86 | 0.975 | 1.17 | 0.043 |
| | ✓ | ✓ | **30.24** | **0.977** | **0.696** | **0.041** |

Table 3: **Supervision granularity ablation.** Comparison of *scene-wise* vs. *object-wise* supervision while toggling the Chamfer term $\mathcal{L}_{\text{CD}}$ and silhouette term $\mathcal{L}_\alpha$.

with object-wise losses leads to improvements across all metrics. Visual fidelity improves, with reconstructions better aligned to the ground truth, while geometric distances decrease substantially. In particular, the large Chamfer Distance observed with scene-wise losses reflects unstable simulation training and inaccurate contact handling, whereas object-wise supervision corrects this, yielding robust optimization and physically meaningful rollouts. In addition, the results demonstrate that single-source supervision is insufficient for robust physical property training.

## 5 DISCUSSION

Despite promising results, our approach has several limitations. It relies on predefined constitutive models and could benefit from directly learning physical models (e.g., via neural networks) to handle materials with unknown properties (Ma et al., 2020; Cao et al., 2024; Zhao et al., 2025). The optimization is computationally intensive and sensitive to initial geometry, motivating more efficient strategies and more robust 3D reconstruction, particularly in cluttered scenes with occlusions. Extending the framework from controlled settings to real-world videos with complex lighting and noise also remains challenging, requiring further efforts to bridge the sim-to-real gap.

## 6 CONCLUSION

We introduce the challenging problem of multi-object system identification from videos and present MOSIV, which serves both as a framework for reconstructing objects' geometry, dynamics, and physical properties and as a comprehensive synthetic dataset for rigorous evaluation. MOSIV reconstructs object-aware Gaussians from multi-view video observations and integrates geometry-driven supervision with a differentiable simulator to recover object geometry, dynamics, and physical properties. This formulation moves beyond prior single-object or discrete material classification methods, enabling physically grounded scene reconstruction and prediction. Extensive experiments on our synthetic benchmark show that MOSIV achieving accurate observable dynamics and strong generalization to future state over complex and diverse scenes.

## ACKNOWLEDGMENT

This work was supported in part by U.S. NSF DBI-2238093. YZ was supported in part by the SoftBank Group–ARM Fellowship.

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

## A    APPENDIX

### A.1    PHYSICAL PARAMETERS.

Our differentiable MPM implementation supports five standard material classes: elastic solids, plasticine, granular media (e.g., sand), Newtonian fluids, and non-Newtonian fluids. For each class, we optimize a small set of physically interpretable parameters:

- *Elasticity*: Young's modulus ($E$) controlling material stiffness, and Poisson's ratio ($\nu$) controlling volume preservation under deformation.
- *Plasticine*: Young's modulus ($E$), Poisson's ratio ($\nu$), and yield stress ($\tau_Y$), which determines the stress level at which permanent (plastic) deformation occurs.
- *Newtonian fluid*: fluid viscosity ($\mu$), governing resistance to velocity changes, and bulk modulus ($\kappa$), governing volume preservation.
- *Non-Newtonian fluid*: shear modulus ($\mu$), bulk modulus ($\kappa$), yield stress ($\tau_Y$), and plastic viscosity ($\eta$), which encodes the decayed, time-dependent resistance to yielding.
- *Sand*: friction angle ($\theta_{\text{fric}}$), which determines the stable slope of a sand pile and controls shear resistance in granular flow.

### A.2    PHYSICAL MODELS

MPM can be paired with a broad family of constitutive and plasticity models. In the continuum formulation, internal forces are expressed via the Cauchy stress $\mathbf{T}$, a tensor field defined as a function of the deformation gradient $\mathbf{F}$. The deformation gradient is tracked on MPM particles to measure their distortion relative to the rest configuration. For plasticity, $\mathbf{F}$ is constrained to an elastic region, and a return mapping $\mathcal{Z}$ projects $\mathbf{F}$ back to the yield surface when this region is violated.

**Elasticity.**    We use a neo-Hookean model for elastic solids. The Cauchy stress is given by

$$J\mathbf{T}(\mathbf{F}) = \mu\left(\mathbf{F}\mathbf{F}^\top\right) + (\lambda\log(J) - \mu)\,\mathbf{I}, \tag{4}$$

where $J = \det(\mathbf{F})$, and $\mu, \lambda$ are Lamé parameters related to Young's modulus $E$ and Poisson's ratio $\nu$ via

$$\mu = \frac{E}{2(1 + \lambda)}, \quad \lambda = \frac{\nu E}{(1 + \nu)(1 - 2\nu)}. \tag{5}$$

**Newtonian fluid.**    We model Newtonian fluids using a $J$-based volumetric term combined with a viscous term:

$$J\mathbf{T}(\mathbf{F}) = \tfrac{1}{2}\mu\left(\nabla\mathbf{v} + \nabla\mathbf{v}^\top\right) + \kappa\left(J - \frac{1}{J^6}\right), \tag{6}$$

where $\mathbf{v}$ is the velocity field, $\mu$ is viscosity, and $\kappa$ is the bulk modulus.

**Plasticine.**    Plasticine is modeled using a St. Venant–Kirchhoff (StVK) elastic model combined with a von Mises plastic return mapping. The elastic stress is

$$J\mathbf{T}(\mathbf{F}) = \mathbf{U}\left(2\mu\boldsymbol{\epsilon} + \lambda\operatorname{Tr}(\boldsymbol{\epsilon})\right)\mathbf{U}^\top, \tag{7}$$

where $\mathbf{F} = \mathbf{U}\boldsymbol{\Sigma}\mathbf{V}^\top$ is the SVD of $\mathbf{F}$ and $\boldsymbol{\epsilon} = \log(\boldsymbol{\Sigma})$ is the Hencky strain.

The von Mises yield condition is

$$\delta\gamma = \|\hat{\boldsymbol{\epsilon}}\| - \frac{\tau_Y}{2\mu} > 0, \tag{8}$$

where $\hat{\boldsymbol{\epsilon}}$ is the deviatoric Hencky strain and $\tau_Y$ is the yield stress. When $\delta\gamma > 0$, the deformation exceeds the elastic region, and the deformation gradient is projected back to the yield surface via the return mapping

$$\mathcal{Z}(\mathbf{F}) = \begin{cases} \mathbf{F} & \delta\gamma \leq 0, \\ \mathbf{U}\exp\!\left(\boldsymbol{\epsilon} - \delta\gamma\frac{\hat{\boldsymbol{\epsilon}}}{\|\hat{\boldsymbol{\epsilon}}\|}\right)\mathbf{V}^\top & \text{otherwise.} \end{cases} \tag{9}$$

**Sand.** Granular materials (sand) are modeled using a Drucker–Prager yield criterion (Klár et al., 2016) with an underlying StVK elastic model. The yielding conditions are

$$\text{tr}(\boldsymbol{\epsilon}) > 0 \quad \text{or} \quad \delta\gamma = \|\hat{\boldsymbol{\epsilon}}\|_F + \alpha \frac{(d\lambda + 2\mu)\,\text{tr}(\boldsymbol{\epsilon})}{2\mu} > 0, \tag{10}$$

where

$$\alpha = \sqrt{\tfrac{2}{3}}\,\frac{2\sin\theta_{\text{fric}}}{3 - \sin\theta_{\text{fric}}}, \tag{11}$$

and $\theta_{\text{fric}}$ is the friction angle. The corresponding return mapping is

$$\mathcal{Z}(\mathbf{F}) = \begin{cases} \mathbf{U}\mathbf{V}^\top & \text{tr}(\boldsymbol{\epsilon}) > 0, \\ \mathbf{F} & \delta\gamma \leq 0,\ \text{tr}(\boldsymbol{\epsilon}) \leq 0, \\ \mathbf{U}\exp\!\big(\boldsymbol{\epsilon} - \delta\gamma\frac{\hat{\boldsymbol{\epsilon}}}{\|\hat{\boldsymbol{\epsilon}}\|}\big)\mathbf{V}^\top & \text{otherwise.} \end{cases} \tag{12}$$

## A.3 GENESIS MULTI-OBJECT DATASET PHYSICAL PARAMETERS

Figs. 7 and 8 shows a sample from the Genesis Multi-Object dataset. We parameterize each material by a set of physical attributes, and assign every object the parameter values associated with its material label. Some of these parameters are selected uniformly at random from a range to increase variation in the dataset, and not all parameters are used by each material. The numerical details are shown below.

| | Young's Modulus $(E)^\dagger$ | Poisson's Ratio $(\nu)^\dagger$ | Density $(\rho)^\dagger$ |
|---|---|---|---|
| Elastic | $[4.75 \times 10^4, 5.25 \times 10^4]$ | $[20, 30]$ | $[800, 1200]$ |
| Elastoplastic | $[4.75 \times 10^4, 5.25 \times 10^4]$ | $[20, 30]$ | $[800, 1200]$ |
| Snow | $[4.75 \times 10^4, 5.25 \times 10^4]$ | $[20, 30]$ | $[800, 1200]$ |
| Liquid | $[4.75 \times 10^4, 5.25 \times 10^4]$ | $[20, 30]$ | $[800, 1200]$ |

| | Shear Modulus $(\mu)^\dagger$ | Yield Stress Range $(\tau_Y)$ | Friction Angle $(\theta)$ |
|---|---|---|---|
| Elastic | – | – | – |
| Elastoplastic | – | $[2.5 \times 10^{-2}, 4.5 \times 10^{-2}]$ | – |
| Snow | – | $[2.5 \times 10^{-2}, 4.5 \times 10^{-2}]$ | – |
| Liquid | $[2.4 \times 10^6, 3.6 \times 10^6]$ | – | – |

Table 4: Physical parameter values for objects in the Genesis Multi-Object Dataset (10 Geometry Shapes)
. † indicates that the value is selected uniformly at random from the given range.

## A.4 MATERIAL POINT METHOD

The Material Point Method (MPM) (Jiang et al., 2016) is a hybrid Eulerian-Lagrangian method for simulation the behavior of continuum materials, based on its physical parameters. MPM represents continuum as particles, each with its own physical parameters, interacting with a background grid from which governing physics laws are solved to obtain an update, and then the updates at grid locations propagate back to the particles. Although (Jiang et al., 2016) covers the mathematical derivations in detail, we provide a brief overview of the method and the most relevant equations.

### A.4.1 INITIALIZATION

The continuum is first represented by a set of discrete particles. We enumerate the particles as $\mathcal{P} = \{1, 2, ..., P\}$ where $P$ is the total number of particles. Each particle $p \in \mathcal{P}$, is initialized with starting position $x_p^0$, velocity $v_p^0$, mass $m_p^0$, volume $V_p^0$, deformation gradient $F_p^0$, and material parameters. Each particle also stores an affine matrix $B_p^0$. The grid is also initialized with grid points $\mathcal{G} = \{1, 2, ..., G\}$ where $G$ is the total number of grid locations.

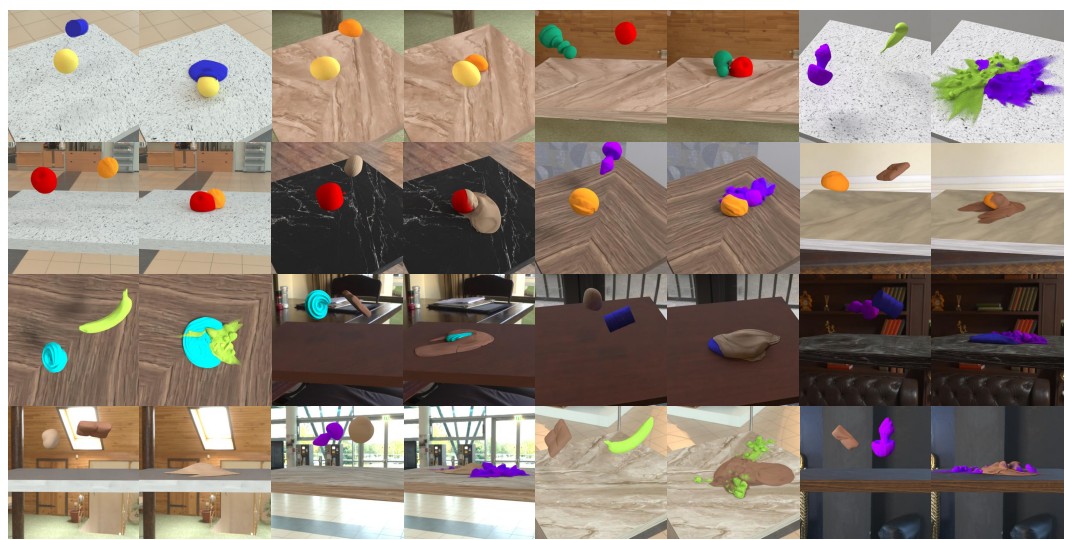

Figure 7: **MOSIV Dataset (2 Objects) Example.**

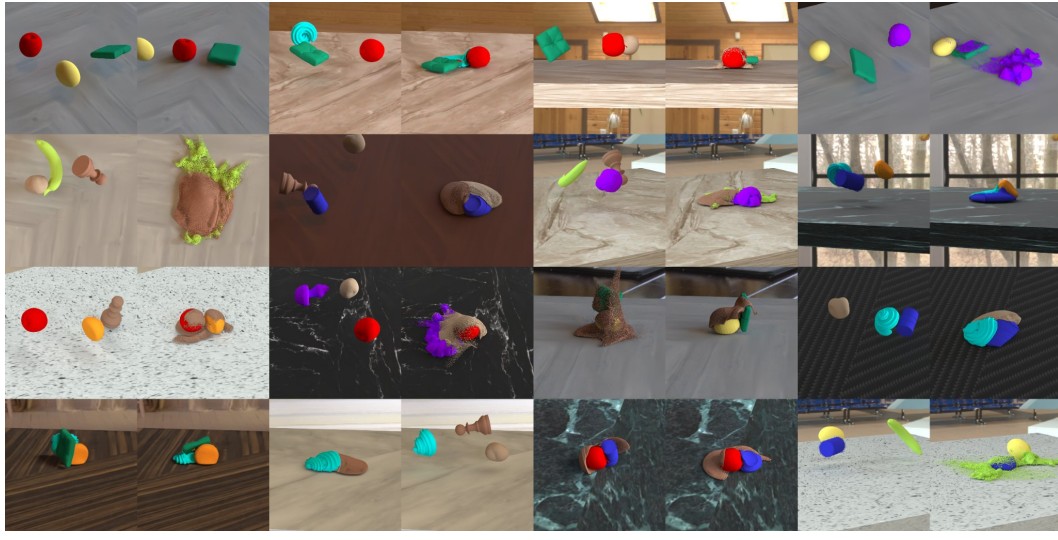

Figure 8: **MOSIV Dataset (3 Objects) Example.**

### A.4.2 PARTICLE TO GRID TRANSFER

At this stage, particle properties are transferred to grid locations to perform physics calculations. Each grid point stores a mass and momentum. At timestep $t$, grid point $i$ has mass:

$$m_i^t = \sum_{p \in \mathcal{P}} w_{ip}^t m_p^t$$

and momentum

$$m_i^t v_i^t = \sum_{p \in \mathcal{P}} w_{ip}^t m_p^t \left( v_p^t + B_p^t (D_p^t)^{-1}(x_i^t - x_p^t) \right)$$

where

$$D_p^t = \sum_{i \in \mathcal{G}} w_{ip}^t (x_i^t - x_p^t)(x_i^t - x_p^t)^\top$$

Grid velocities are calculated by dividing momentum by mass. In the case that the mass is zero, the velocity is instead set to 0.

### A.4.3 COMPUTING GRID FORCES AND VELOCITY UPDATE

Now, the force acting upon each grid point as a result of elastic stresses from nearby particles is calculated for grid point $i$ at timestep $t$ as:

$$f_i^t = -\sum_{p \in \mathcal{P}} V_p^0 \left( \frac{\partial \Psi_p}{\partial F} (F_p^t) \right) (F_p^t)^\top \nabla w_{ip}^t$$

Here, $F_p^t$ is the deformation gradient at time $t$ for particle $p$ and $\frac{\partial \Psi_p}{\partial F}$ represents the first Piola-Kirchhoff stress tensor at that particle. Now, we can update the velocity at each grid point with:

$$v_i^{t+1} = v_i t + \Delta t f_i^t(x_i^t)/m_i$$

In this step, boundary conditions and collisions are also taken into account and resolved.

### A.4.4 GRID TO PARTICLE TRANSFER

Now that the relevant values have been computed at the grid points, we can propagate these changes back to the particles by updating their deformation gradients as follows for particle $p$ at timestep $t$:

$$F_p^{t+1} = \left( \mathbf{I} + \Delta t \sum_{i \in \mathcal{G}} v_i^{t+1} (\nabla w_{ip}^t)^\top \right) F_p^t$$

At this stage, updates to each particle's velocity and $B$ affine matrix are also calculated. The velocity is calculated as:

$$v_p^{t+1} = \sum_{i \in \mathcal{G}} w_{ip}^t v_i^t$$

The affine matrix is updated as:

$$B_p^{t+1} = \sum_{i \in \mathcal{G}} w_{ip}^t v_i^t (x_i^t - x_p^t)^\top$$

### A.4.5 PARTICLE UPDATE

Finally, with the updated velocities, the particle locations can now also be updated as follows for particle $p$ at timestep $t$:

$$x_p^{t+1} = x_p^t + \Delta t v_p^{t+1}$$

Finally, with the updated velocities, the particle locations can now also be updated as follows for particle $p$ at timestep $t$:

$$x_p^{t+1} = x_p^t + \Delta t v_p^{t+1}$$

### A.5 MORE QUANTITATIVE RESULTS ON MOSIV DATASET

In Tab. 5, we report an average performance comparison of MOSIV and all baselines on the MOSIV dataset. Across all reported metrics, MOSIV consistently outperforms both OPGS and CoupNeRF*, indicating sharper, more structurally faithful reconstructions and more accurate geometric distributions. These gains persist in the both observable and future regime, underscoring robust system identification that generalizes beyond observed frames.

| Method | Observable Simulation | | | | Future Simulation | | | |
|---|---|---|---|---|---|---|---|---|
| | PSNR ↑ | SSIM ↑ | CD ↓ | EMD ↓ | PSNR ↑ | SSIM ↑ | CD ↓ | EMD ↓ |
| OPGS(Lin et al., 2025) | 25.93 | 0.945 | 11.79 | 0.095 | 19.00 | 0.888 | 51.92 | 0.199 |
| CoupNeRF*(Li et al., 2024a) | 25.88 | 0.962 | 0.927 | 0.047 | 20.62 | 0.942 | 1.859 | 0.062 |
| MOSIV (Ours) | **31.17** | **0.975** | **0.389** | **0.033** | **28.92** | **0.964** | **0.894** | **0.050** |

Table 5: **Avg. Performance Comparison on MOSIV dataset.** * for reproduced implementation.

## A.6 COMPUTATION OVERHEAD COMPARISON

For a single 30-frame sequence (same resolution and views as in our main experiments), we measure the average training time and peak GPU memory. Specifically, MOSIV and CoupNeRF are both trained on a single NVIDIA A6000 (48 GB). In contrast, OPGS cannot be trained on an A6000 due to out-of-memory (OOM) and therefore requires an NVIDIA H100 (80 GB). As shown in Tab. 6, despite running on a less powerful GPU, MOSIV consitently achieves the lowest runtime and peak memory.

| Metric \ Method | CoupNeRF* | OPGS | MOSIV |
|---|---|---|---|
| Training Time (s) ↓ | 9591.09 | 5263.70 | 5021.46 |
| Peak GPU Memory (GB) ↓ | 31.14 | 61.08 | 29.79 |

Table 6: Runtime and memory cost

## A.7 EXPERIMENT RESULTS ON MOSIV DATASET EXTENSION (3 OBJECTS)

We augment MOSIV with a three-object interaction benchmark to further stress complex, contact-rich dynamics. As shown in Tab. 7, across this more challenging setting, our approach demonstrates consistently strong quantitative results.

| Method | Observable Simulation | | | | Future Simulation | | | |
|---|---|---|---|---|---|---|---|---|
| | PSNR ↑ | SSIM ↑ | CD ↓ | EMD ↓ | PSNR ↑ | SSIM ↑ | CD ↓ | EMD ↓ |
| MOSIV | 23.98 | 0.941 | 2.401 | 0.043 | 19.81 | 0.904 | 5.800 | 0.095 |

Table 7: **Quantitative Evaluation of MOSIV Dataset Extension**

## A.8 SENSITIVITY ANALYSIS

We assess robustness to reconstruction inaccuracy by perturbing Gaussians trained from fixed ground-truth point clouds before system identification. For each object, we add Gaussian noise with standard deviation $\sigma = \alpha \cdot (\text{xyz}_{\max} - \text{xyz}_{\min})$ where $\alpha$ is the noise magnitude shown in Tab. 8 and Tab. 9. This scales the perturbation relative to each object's spatial extent. For every frame, we consider two variants of corruption: (1) *i.i.d. noise*, where each point receives an independent sample, and (2) *shared noise*, where all points of an object share the same offset. We then apply MOSIV for system identification and long-horizon prediction. As shown in the tables, increasing the noise level from $\alpha = 0.005$ to $\alpha = 0.02$ produces a moderate decrease in PSNR (about 1–2 dB) and a corresponding increase in CD/EMD, while SSIM remains high ($\geq 0.95$). At the largest perturbation level ($\alpha = 0.05$), PSNR and SSIM remain in the 23–24 dB and 0.93–0.94 ranges, respectively, and CD/EMD increase further but stay within the same order of magnitude. Overall, the metrics degrade smoothly rather than abruptly, indicating that MOSIV can tolerate substantial reconstruction noise.

| Noise level($\alpha$) | Observable Simulation | | | | Future Simulation | | | |
|---|---|---|---|---|---|---|---|---|
| | PSNR ↑ | SSIM ↑ | CD ↓ | EMD ↓ | PSNR ↑ | SSIM ↑ | CD ↓ | EMD ↓ |
| 0.005 | 27.86 | 0.96 | 0.22 | 0.03 | 27.47 | 0.96 | 0.29 | 0.03 |
| 0.020 | 26.06 | 0.95 | 0.41 | 0.03 | 25.43 | 0.95 | 0.52 | 0.05 |
| 0.050 | 23.73 | 0.94 | 1.27 | 0.05 | 23.39 | 0.93 | 1.72 | 0.07 |

Table 8: **Sensivity analysis of i.i.d. noise**

| Noise level ($\alpha$) | Observable Simulation | | | | Future Simulation | | | |
|---|---|---|---|---|---|---|---|---|
| | PSNR ↑ | SSIM ↑ | CD ↓ | EMD ↓ | PSNR ↑ | SSIM ↑ | CD ↓ | EMD ↓ |
| 0.005 | 27.82 | 0.96 | 0.24 | 0.03 | 27.43 | 0.96 | 0.35 | 0.03 |
| 0.020 | 26.17 | 0.95 | 0.38 | 0.03 | 25.57 | 0.95 | 0.49 | 0.04 |
| 0.050 | 23.92 | 0.93 | 1.14 | 0.04 | 23.52 | 0.93 | 1.50 | 0.06 |

Table 9: **Sensitivity analysis of shared noise**

## A.9 MORE QUALITATIVE RESULTS

We conduct extra qualitative comparison of multi-material interaction, showing the results of OmniphysGS (Lin et al., 2025), OmniphysGS w/ Oracle (Lin et al., 2025), and our MOSIV in Figs. 9 to 18. Each figure compares the results of both observed and predicted frames and MOSIV shows finer deformations with greater accuracy, demonstrating superior adaptability to diverse materials interactions.

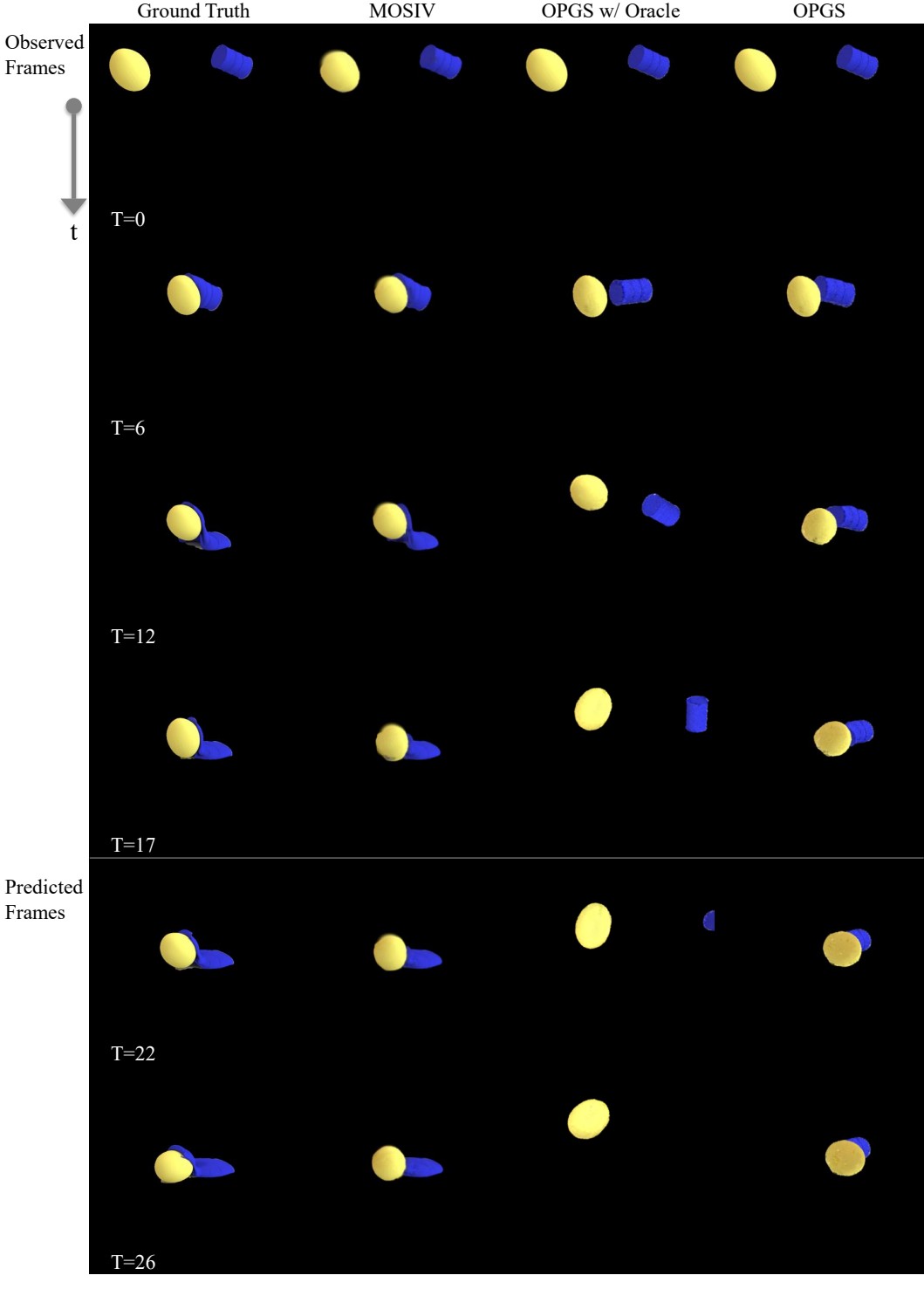

Figure 9: **Qualitative Comparison with material type elastic/fluid (E/F).**

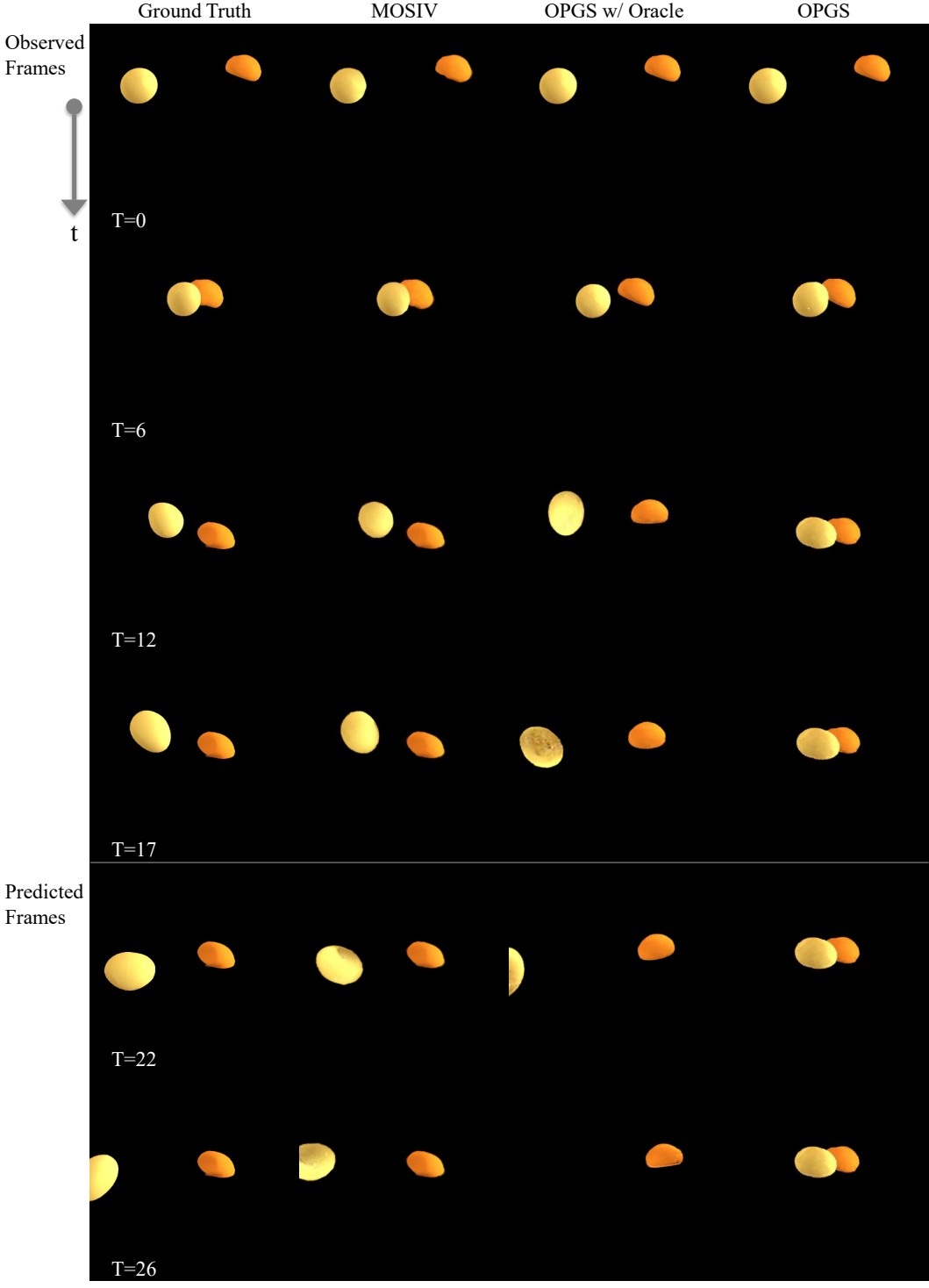

Figure 10: **Qualitative Comparison with material type elastic/plasticine (E/P).**

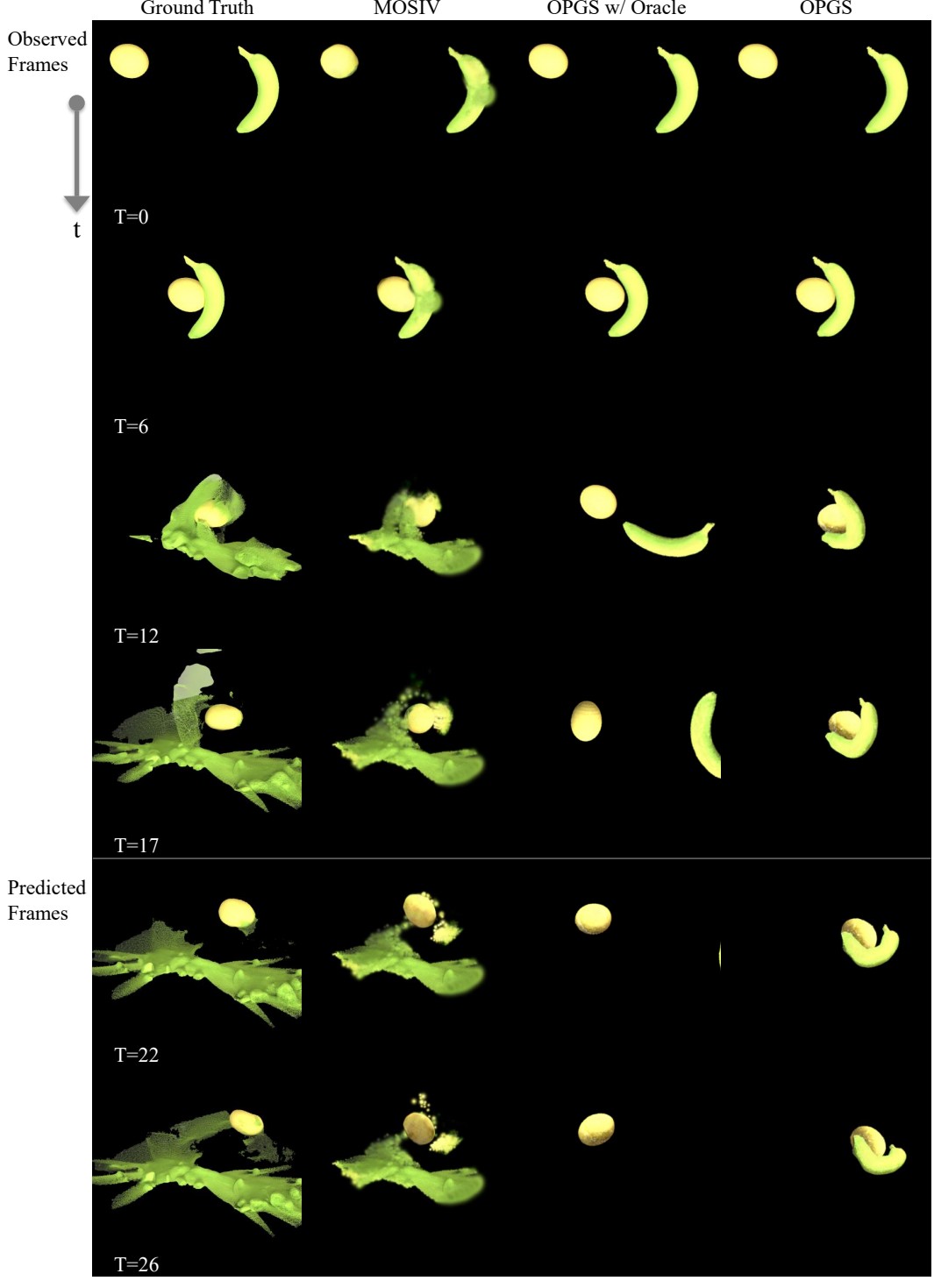

Figure 11: **Qualitative Comparison with material type elastic/sand (E/S).**

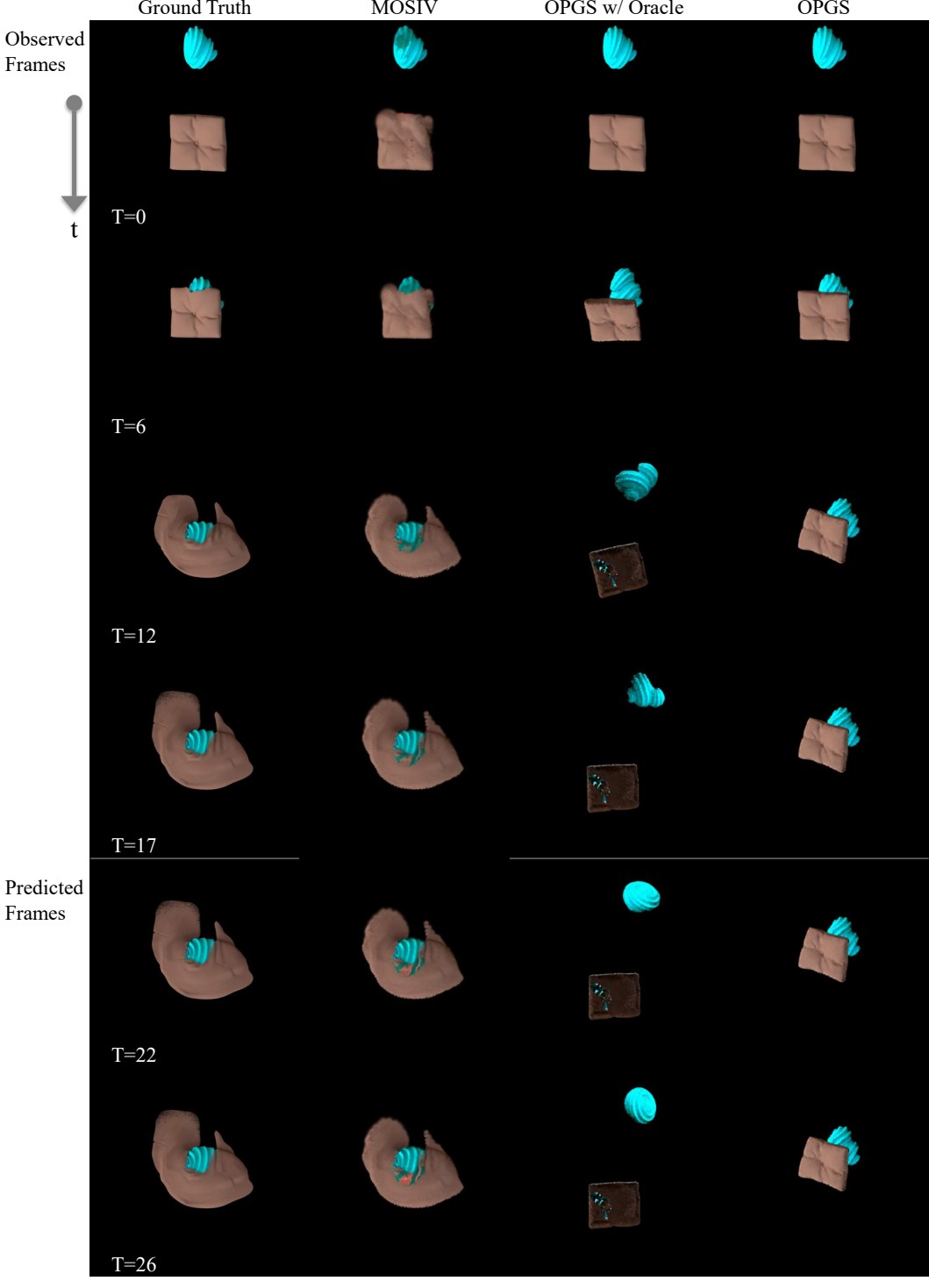

Figure 12: **Qualitative Comparison with material type fluid/sand (F/S).**

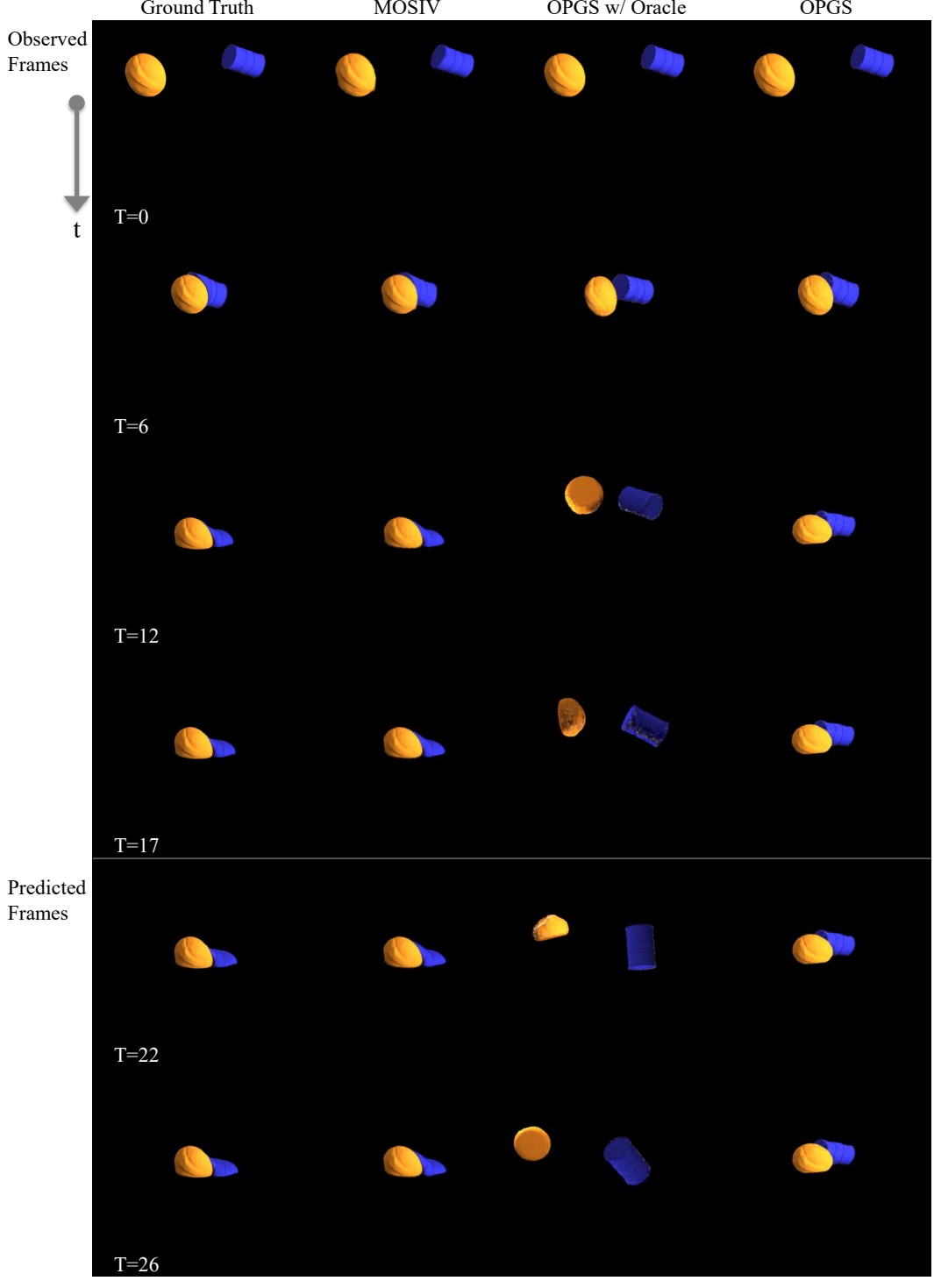

Figure 13: **Qualitative Comparison with material type plasticine/fluid (P/F).**

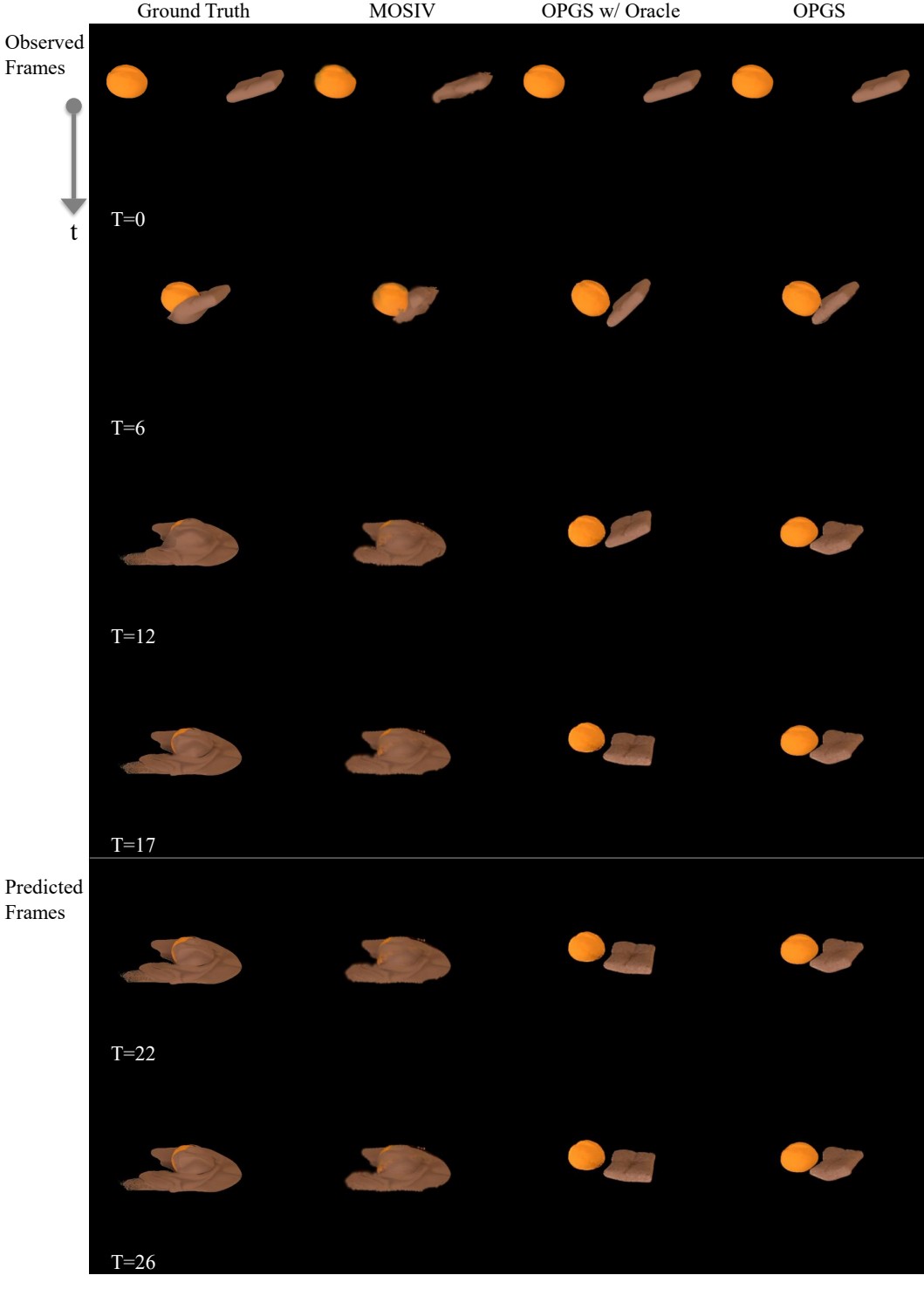

Figure 14: **Qualitative Comparison with material type sand/plasticine (S/P).**

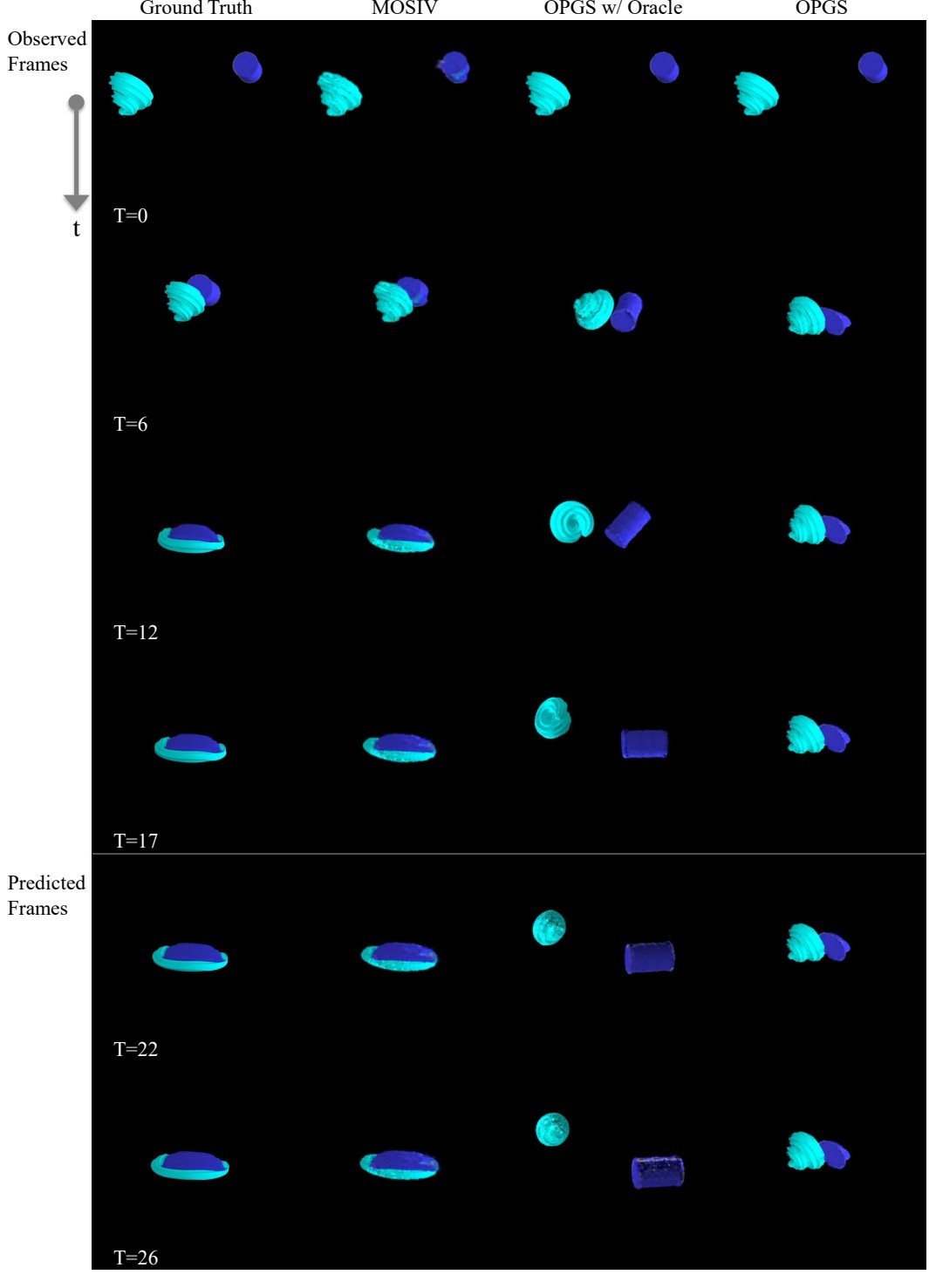

Figure 15: **Qualitative Comparison with material type fluid/fluid (F/F).**

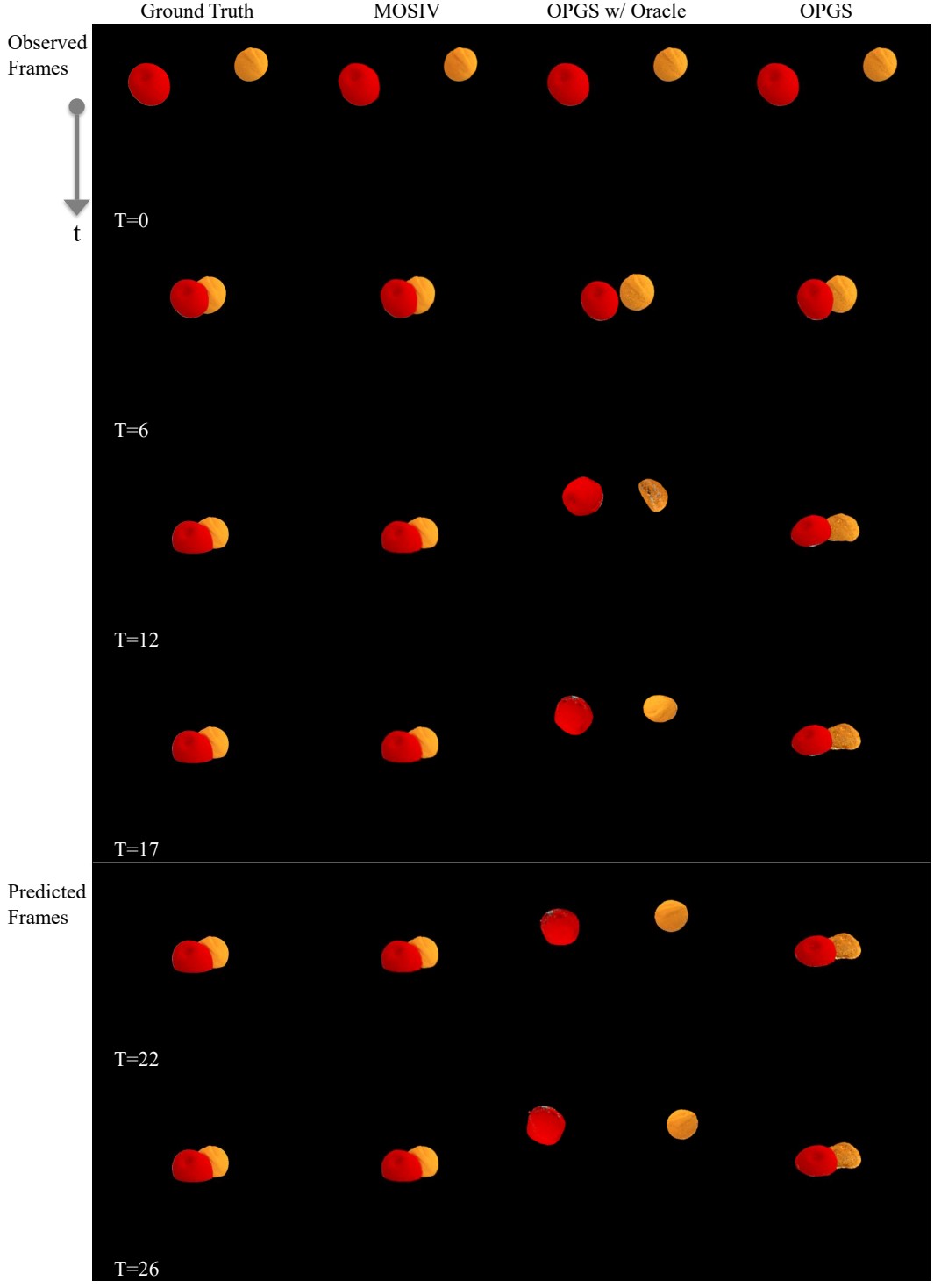

Figure 16: **Qualitative Comparison with material type plasticine/plasticine (P/P).**

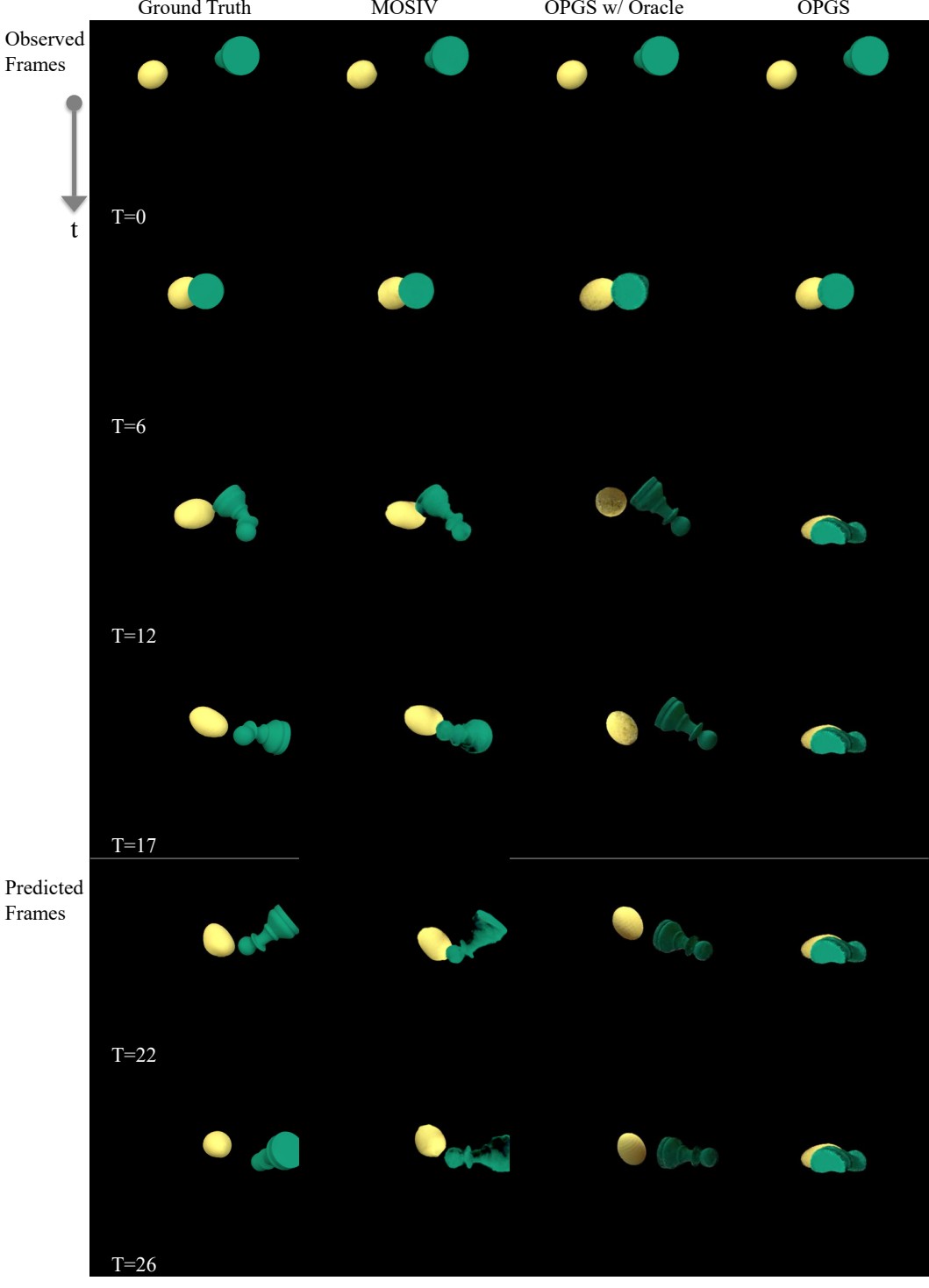

Figure 17: **Qualitative Comparison with material type elastic/elastic (E/E).**

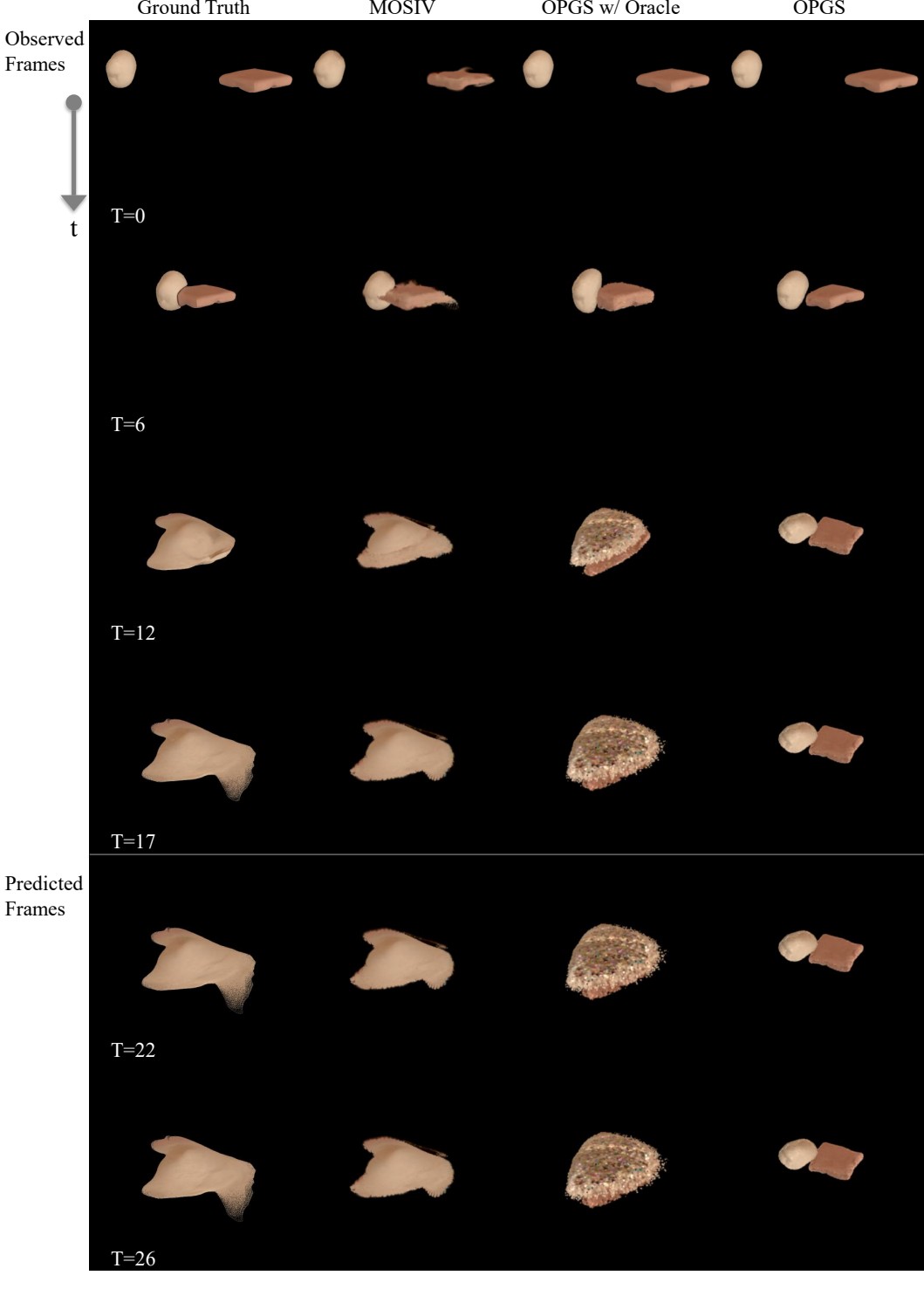

Figure 18: **Qualitative Comparison with material type sand/sand (S/S).**

