# OpenReview forum: "Multi-Object System Identification from Videos"
_ICLR.cc/2026/Conference — ICLR 2026 Poster_

### Official Review · Reviewer_SAcW · 2025-10-31

**Soundness:** 3
**Presentation:** 3
**Contribution:** 3
**Rating:** 6
**Confidence:** 3

**Summary:**

The paper introduces a synthetic video dataset that presents contact and interaction between different materials. A baseline framework that identifies the continuous, object-specific physical properties is also proposed, with geometric-driven supervision and object-aware dynamic component.

**Strengths:**

- I found this paper reused and properly directed to external sources for development that helps the readers from diverse backgrounds.
- A few features I see interesting to have together in this framework: simulation-ready continuum for multiple objects across all time, permaterial parameters to predict future motions.

**Weaknesses:**

My background is not in physics simulation, so I look at the paper from the perspective of a computer vision researcher.
- It is unclear how long the predicted horizon is. The example videos appear quite short—only a few seconds in duration, and seem not to show motion prediction (which I find different from trajectory prediction. Trajectory prediction is accumulating past locations, while motion prediction is to predict future locations). Please clarify the temporal length of the predictions.
- How many object categories are supported in the dataset? A summary table presenting dataset statistics would be helpful for readers. Also, I find that only two object interactions are limited compared to existing datasets such as CLEVRER.
- Additionally, please discuss or evaluate the model's ability to generalize to out-of-distribution in both object categories and interactions. This would also help to strengthen section subsection 3.7.

**Questions:**

See weaknesses.

---

> ### Author Response · Authors · 2025-11-26
>
> > Please clarify the temporal length of the predictions.
>
> Our dataset sequences contain 30 frames in total (recorded at 24 fps). Of these, the first 20 frames are used as observed frames, while the remaining 10 frames are reserved for future-state prediction. The predicted segment therefore spans approximately 0.4 s of unseen dynamics, covering the post-contact phase after collision and deformation.
>
> > How many object categories are supported in the dataset?
>
> Our dataset includes 10 object geometry (egg, pawn, apple, bread, cream, barrel, potato, cushion, banana, and mushroom) and 4 material types (elastic, elastoplastic, liquid, sand), resulting in 45 multi-view interaction sequences covering diverse geometry–material combinations. Additionally, we have provided a comprehensive illustration of the consitutive models in Appendix A.1 of the updated manuscript. We utilize four standard differentiable constitutive models, neo-Hookean elasticity, StVK plasticity, Newtonian fluid, and Drucker–Prager sand. This ensures that our task captures a diverse range of physical dynamics.
>
> > I find that only two object interactions are limited compared to existing datasets such as CLEVRER.
>
> We adopt the two-object interaction to incorporate more baseline comparison, as more objects will add up the runtime as well as memory consumption. To demonstrate the flexibility of our method, we have generated a three-object interaction benchmark to evaluate OPGS, CoupNeRF, as well as our MOSIV in this more challenging setting, for which the quantitative and qualitative results have been updated in the revised manuscript. This shows the flexibility of our method. OPGS and CoupNeRF both resulted in out-of-memory (OOM). OmniPhysGS formulates the scene with hundreds of point groups, each with its own set of physical parameters that are predicted and then jointly optimized through MPM. This per-group parameterization makes both the state and gradient storage scale with the number of groups, leading to very high memory usage. By comparison, MOSIV uses a compact per-object parameter vector (one set of material parameters per object), which keeps the optimization problem and memory footprint manageable even as we increase the number of interacting objects.
>
>
>
> | **Method** | **PSNR ↑** | **SSIM ↑** | **CD ↓** | **EMD ↓** | **PSNR ↑** | **SSIM ↑** | **CD ↓** | **EMD ↓** |
> |------------|-------------|-------------|-----------|------------|-------------|-------------|-----------|------------|
> | **MOSIV**  | 23.98       | 0.941       | 2.401     | 0.043      | 19.81       | 0.904       | 5.800     | 0.095      |
>
> **Table R1. Observable (Left) and Future (Right) state simulation on MOSIV three-object dataset.**
>
> > Additionally, please discuss or evaluate the model's ability to generalize to out-of-distribution in both object categories and interactions.
>
> In Fig. 3, we investigate the novel interaction of switching object material types. In the first column, the original simulation captures elastic egg vs. elastoplastic bread, and the novel simulation turns to elastoplastic egg vs. elastic bread. The second column shows elastic egg vs. liquid barrel and liquid egg vs. elastic barrel. The third column demonstrates elastoplastic apple vs. sand cream and sand apple vs. elastoplastic cream. The resulting simulation generates plausible, novel collision behaviors—such as plastic deformation in the egg or elastic rebound in the bread that were never observed in the input video. This confirms that our model successfully disentangles physical parameters from specific object geometries, enabling generalization to novel object-material combinations and interaction dynamics (e.g., elastoplastic apple vs. sand cream) beyond the training distribution. We have updated the discussion accordingly in the revised manuscript.

---

> > ### Author Response · Authors · 2025-11-28
> > **Have we addressed all the issues?**
> >
> > Dear Reviewer,
> >
> > Thank you again for your thoughtful feedback. We have carefully addressed all the comments and questions raised in your earlier review. As the discussion phase is nearing its end, we would greatly appreciate it if you could let us know whether our responses satisfactorily resolve the issues you identified.
> >
> > We sincerely appreciate your consideration of a potential rating update once all concerns have been addressed.
> >
> > Thank you for your time and effort.

---

### Official Review · Reviewer_yotJ · 2025-11-01

**Soundness:** 4
**Presentation:** 2
**Contribution:** 4
**Rating:** 6
**Confidence:** 4

**Summary:**

The paper presents MOSIV (Multi-Object System Identification from Videos), a novel framework designed to infer continuous physical constitutive parameters for multiple interacting objects directly from multi-view video observations. By employing per-object geometric reconstruction followed by the system identification of continuous parameters, MOSIV moves beyond prior work that relies solely on selecting from a fixed, categorical library of expert constitutive models. This advancement significantly enhances the fidelity, precision, and physical plausibility of subsequent dynamic simulations.

**Strengths:**

+ Pioneering Continuous Parameter Identification: The primary strength is the innovative shift from categorical material classification to the estimation of continuous constitutive parameters. This allows for highly granular, object-specific physical calibration, overcoming a major limitation of current video-to-physics pipelines.
+ Effective Multi-Object Handling: The framework successfully tackles the complexity of simultaneous system identification across multiple objects that are undergoing complex interactions (e.g., contact, collision) within the same observed scene, which is a critical capability for real-world robotics and simulation.
+ Robust Framework Integration: The successful integration of geometric reconstruction and a differentiable physics pipeline suggests a robust, end-to-end optimization strategy capable of generating strongly calibrated and faithful physical models from raw visual data.

**Weaknesses:**

- Overall, this paper presents a clear goal and a coherent overall narrative. However, the main issue lies in the lack of clear motivation behind many of the detailed design choices. In several instances, important implementation details are missing, which can lead to reader confusion and make the paper difficult to follow.
- According to the description in subsection 3.3, objects are first represented as Gaussians, then converted into voxels, and finally transformed into particles. However, the whole process is quite unclear given the current simple explanation. Moreover, what parameters define these particles? Are they simply points? If so, why not use point clouds as the initial representation instead?
- In subsection 3.5, why are silhouettes and surfaces used as the objective? Is this a common approach for such tasks? If so, please provide appropriate references; if not, please clarify the rationale behind this choice.
- In Figure 4, different methods appear to have different observed frames. Does this imply that they were simulated using different parameters or settings? Shouldn’t the observed frames be consistent across methods to ensure a fair comparison?
- Given that the method involves per-object geometric reconstruction and the iterative optimization of continuous parameters, the computational expense is likely very high. Could the authors provide a rigorous analysis of the computational complexity (e.g., model size, training time, and inference time) and a comparison with other methods?
- How robust is MOSIV to different levels of reconstruction uncertainty? Did the authors conduct a sensitivity analysis showing how small errors in the initial reconstruction propagate into the estimated values of the constitutive parameters?
- While the parameters are continuous, the underlying physics model must still be chosen. Does MOSIV include a mechanism for a priori selecting the correct base constitutive model type, or is that externally provided?

**Questions:**

See weaknesses

---

> ### Author Response · Authors · 2025-11-26
>
> > the main issue lies in the lack of clear motivation behind many of the detailed design choices
>
> We are sorry for the confusion. If possible, could you specify the detailed design choices? We are more than happy to improve our updated manuscript.
>
> > objects are first represented as Gaussians, then converted into voxels, and finally transformed into particles. However, the whole process is quite unclear given the current simple explanation. Moreover, what parameters define these particles? Are they simply points? If so, why not use point clouds as the initial representation instead?
>
> The three-stage representation is adopted as different stages of our pipeline need different properties:
>
> - **Gaussians (for dynamic reconstruction).**
>   We first reconstruct the object as 4D Gaussians. Compared to point clouds, Gaussians support high-quality inverse rendering and handle occlusion and appearance changes due to uneven illumination well.
>
> - **Voxels (for internal filling).**
>   From these Gaussians we build a per-object occupancy volume. This voxelization step is used only to perform internal filling and surface extraction. It gives us intermediate representation towards a dense interior, a well-defined surface for geometry supervision, and a meaningful continuum for system identification by differentiable simulation.
>
> - **Particles (for MPM simulation).**
>   Finally, we sample points from the filled volume to obtain MPM particles. These particles are not only geometric points. Each particle carries a 3D position, a material type label (elastic, plasticine, sand, etc.), and the associated physical properties, e.g., $E,\nu,\mu,\kappa,\tau_Y,\eta,\theta_{\mathrm{fric}}$, shared within that object, as well as MPM particle states like affine momemtum.
>
> Hence, the three-stage pipeline is necessary to bridge video-based dynamic reconstruction and and simulation for systyem identification.
>
> > why are silhouettes and surfaces used as the objective? Is this a common approach for such tasks?
>
> The 3D geometric and 2D photometic supervisions are a common practice, as used in GIC and MASIV. They complements each other. Surface alignment is critical for accurate contact and collision resolution, while multi-view silhouettes provide strong, differentiable gradients to correct global shape and deformation.
>
> > Shouldn’t the observed frames be consistent across methods to ensure a fair comparison?
>
> We ensure strict fairness: all methods utilize identical initial conditions and observed frames. The visual discrepancies in Figure 4 arise not from experimental inconsistency, but solely from the baseline’s (OPGS) failure to learn accurate dynamics. This error causes significant simulation drift at the specific timestamps where our method remains robust.
>
> > Could the authors provide a rigorous analysis of the computational complexity (e.g., model size, training time, and inference time) and a comparison with other methods?
>
> | **Metric \\ Method**        | **CoupNeRF\*** | **OPGS** | **MOSIV** |
> |-----------------------------|----------------|----------|-----------|
> | **Training Time (s) ↓**     | 9591.09        | 5263.70  | 5021.46   |
> | **Peak GPU Memory (GB) ↓**  | 31.14          | 61.08    | 29.79     |
>
> **Table R1. Traning Time and GPU Memory Comparison.**

---

> ### Author Response · Authors · 2025-11-26
>
> > Did the authors conduct a sensitivity analysis showing how small errors in the initial reconstruction propagate into the estimated values of the constitutive parameters?
>
> We updated a sensitivity analysis by perturbing the reconstructed Gaussians before lifting them to particles. Concretely, we add i.i.d. Gaussian noise of magnitude $\sigma \in \{0.005, 0.02, 0.05\}$ to the reconstructed Gaussian centers and then rerun MOSIV for system identification only. Here $\sigma$ denotes the standard deviation, which are used to generate Gaussian noise, multiply by the longest length of the object Gaussian bounding box, and then apply as the perturbation. Tables R2–R3 report the resulting metrics in observed state simulation and future state simulation. For small perturbations ($\alpha \le 0.02$), PSNR/SSIM change only slightly and CD/EMD remain essentially unchanged, indicating that moderate reconstruction errors have little effect on the calibrated parameters and long-horizon rollouts. Only for the largest perturbation ($\alpha = 0.05$) do we observe noticeable degradation, as expected.
>
> | **Noise std $\sigma$** | **PSNR ↑** | **SSIM ↑** | **CD ↓** | **EMD ↓** |
> |-------------|------------------|------------------|----------------|-----------------|
> | **0.005**   | 21.06            | 0.95             | 0.32           | 0.03            |
> | **0.020**   | 20.43            | 0.94             | 0.32           | 0.03            |
> | **0.050**   | 16.91            | 0.90             | 1.26           | 0.05            |
>
> **Table R2. Sensitivity to reconstructed Gaussian perturbations in observed state simulation**
>
> | **Noise std $\sigma$** | **PSNR ↑** | **SSIM ↑** | **CD ↓** | **EMD ↓** |
> |-------------|-------------------|-------------------|-----------------|------------------|
> | **0.005**   | 18.36             | 0.93              | 2.89            | 0.07             |
> | **0.020**   | 18.21             | 0.93              | 3.72            | 0.09             |
> | **0.050**   | 15.75             | 0.88              | 21.07           | 0.16             |
>
> **Table R3. Sensitivity to initial Gaussian perturbations in future state simulation**
>
>
> > Does MOSIV include a mechanism for a priori selecting the correct base constitutive model type, or is that externally provided?
>
> We provide 2D material type segmentation masks and optimize per-Gaussian material type during dynamic reconstruction. This optimized material type determines the constitutive model used for simulation.

---

> ### Author Response · Authors · 2025-11-28
> **Have we addressed all the issues?**
>
> Dear Reviewer,
>
> Thank you again for your thoughtful feedback. We have carefully addressed all the comments and questions raised in your earlier review. As the discussion phase is nearing its end, we would greatly appreciate it if you could let us know whether our responses satisfactorily resolve the issues you identified.
>
> We sincerely appreciate your consideration of a potential rating update once all concerns have been addressed.
>
> Thank you for your time and effort.

---

### Official Review · Reviewer_yKeE · 2025-11-01

**Soundness:** 2
**Presentation:** 3
**Contribution:** 3
**Rating:** 6
**Confidence:** 2

**Summary:**

This paper introduces MOSIV, a new framework created to solve the problem of identifying the physical properties of multiple interacting objects simultaneously from a video. MOSIV works by using a differentiable simulator. It directly optimizes the specific, continuous material parameters for each object by trying to match the geometry observed in the video. MOSIV also presents a new synthetic benchmark with contact-rich, multi-object interactions.

**Strengths:**

- the vision of learning physical properties and its interactions directly from videos seems appealing
- The overall approach is logical and builds upon several state-of-the-art components, including the dynamic Gaussian Splatting for reconstruction and a differentiable MPM for physics-based parameter identification
- the author introduces a new multi-object dataset with diverse geometry, materials properties and physical motions, that could be used from the community

**Weaknesses:**

- the paper studies multi-object system interaction, but the proposed dataset only contains two-object interactions, also 30 frames of interactions seems quite short for evaluation, given the authors claim that the calibrated models generalizes to "long-horizon predictions of complex multi-object dynamics"

**Questions:**

- The method appears to be computationally expensive? It involves (1) optimizing a 4D Gaussian scene and assigning instance partitions, (2) converting object's reconstruction into simulation-ready continuum (3) running a differentiable MPM simulation to optimize per-object parameter vectors, and (4) optimizing this entire unrolled simulation. What are the typical run time and memory footprint? how does it compare with baseline methods?
- what are the novel interactions in section 3.7? if my understanding is correctly, the physical motions are kept the same, you only change the materials?
- what consitutive models have been used? how are they chosen, can you provide more details on that?

---

> ### Author Response · Authors · 2025-11-26
>
> > but the proposed dataset only contains two-object interactions
>
> We have generated a three-object interaction benchmark to evaluate OPGS, CoupNeRF, as well as our MOSIV in this more challenging setting, for which the quantitative results have been updated in the revised manuscript. This shows the flexibility of our method. OPGS and CoupNeRF both resulted in out-of-memory (OOM). OmniPhysGS formulates the scene with hundreds of point groups, each with its own set of physical parameters that are predicted and then jointly optimized through MPM. This per-group parameterization makes both the state and gradient storage scale with the number of groups, leading to very high memory usage. By comparison, MOSIV uses a compact per-object parameter vector (one set of material parameters per object), which keeps the optimization problem and memory footprint manageable even as we increase the number of interacting objects.
>
>
> | **Method** | **PSNR ↑** | **SSIM ↑** | **CD ↓** | **EMD ↓** | **PSNR ↑** | **SSIM ↑** | **CD ↓** | **EMD ↓** |
> |------------|-------------|-------------|-----------|------------|-------------|-------------|-----------|------------|
> | **MOSIV**  | 23.98       | 0.941       | 2.401     | 0.043      | 19.81       | 0.904       | 5.800     | 0.095      |
>
> **Table R1. Observable (Left) and Future (Right) state simulation on MOSIV three-object dataset.**
>
> > also 30 frames of interactions seems quite short for evaluation
>
> Our 30-frame sequences are intentionally designed to capture the complete interaction phase, from pre-contact through peak deformation to stabilization, omitting only uninformative post-contact rigid drift. This duration is also a common practice, roughly doubling the temporal extent of datasets like PAC-NeRF (14 frames), and remaining the same as Spring-Gaus (30 frames). Crucially, our claim about “long-horizon predictions” is evaluated not on the observed time window alone, but through future state simulation and novel interaction (Sec.
> , Tab. 2, Fig. 3–5). We fit parameters using the observed interaction, then explicitly test their stability by rolling out the simulator to predict unseen future frames and novel interaction scenarios.
>
> > The method appears to be computationally expensive? ... What are the typical run time and memory footprint? how does it compare with baseline methods?
>
> | **Metric \\ Method**        | **CoupNeRF\*** | **OPGS** | **MOSIV** |
> |-----------------------------|----------------|----------|-----------|
> | **Training Time (s) ↓**     | 9591.09        | 5263.70  | 5021.46   |
> | **Peak GPU Memory (GB) ↓**  | 31.14          | 61.08    | 29.79     |
>
> **Table R2. Training Time and GPU Memory Comparison**
>
>
> We have added a runtime/memory comparison in the revised manuscript. For a single 30-frame sequence (same resolution and views as in our main experiments), we measure the average training time and peak GPU memory. Specifically, MOSIV and CoupNeRF$^*$ are both trained on a single NVIDIA A6000 (48 GB). In contrast, OPGS cannot be trained on an A6000 due to out-of-memory (OOM) and therefore requires an NVIDIA H100 (80 GB). Despite running on a less powerful GPU, MOSIV consitently achieves the lowest runtime and peak memory. This shows that our MOSIV is the most time and memory efficient method compared to existing works.
>
> > what are the novel interactions in section 3.7? if my understanding is correctly, the physical motions are kept the same, you only change the materials?
>
> Your understanding is correct. After dynamic reconstruction, we get the initial state of an object comprising initial velocity, particle position, etc. Upon this same initial state, we can modify the object material, initial pose, initial velocity, force field, etc. All these changes will all result in novel motions. Since existing datasets already span diverse initial velocity and pose, we focus on the novel interaction effect brought by changing materials.
>
>
> > what consitutive models have been used? how are they chosen, can you provide more details on that?
>
> We have provided a comprehensive illustration of the consitutive models in Appendix A.1 of the updated manuscript. We utilize four standard differentiable constitutive models, neo-Hookean elasticity, StVK plasticity, Newtonian fluid, and Drucker–Prager sand. This ensures that our task captures a diverse range of physical dynamics.

---

> ### Author Response · Authors · 2025-11-28
> **Have we addressed all the issues?**
>
> Dear Reviewer,
>
> Thank you again for your thoughtful feedback. We have carefully addressed all the comments and questions raised in your earlier review. As the discussion phase is nearing its end, we would greatly appreciate it if you could let us know whether our responses satisfactorily resolve the issues you identified.
>
> We sincerely appreciate your consideration of a potential rating update once all concerns have been addressed.
>
> Thank you for your time and effort.

---

### Official Review · Reviewer_dfDn · 2025-11-02

**Soundness:** 2
**Presentation:** 2
**Contribution:** 1
**Rating:** 2
**Confidence:** 5

**Summary:**

The paper proposes a framework to recover material properties from multi-view videos of multi-object scenes. The task setting is similar to PAC-NeRF except that multiple objects exist in the same scene. In additional to pixel supervision, the method also reconstructs particle trajectories using a 4D Gaussian framework, so that 3D surface loss can also be used as supervision.

**Strengths:**

- The experiments are comprehensive and promising.

**Weaknesses:**

- The contribution is limited. The following two works are not cited. Justification of contributions needs revision.
    - The task of multi-object system identification from videos is not novel: [1] also tackles multi-object system ID based on NeRF.
    - Geometry-driven supervision is not novel: [2] uses 4D Gaussian to reconstruction mesh sequence from multiview videos and use 3D loss to tune physical parameters of cloth.

[1] Li, J., Gao, Y., Song, W., Li, Y., Li, S., Hao, A. and Qin, H., 2024, October. CoupNeRF: Property‐aware Neural Radiance Fields for Multi‐Material Coupled Scenario Reconstruction. In Computer Graphics Forum (Vol. 43, No. 7, p. e15208).

[2] Zheng, Y., Zhao, Q., Yang, G., Yifan, W., Xiang, D., Dubost, F., Lagun, D., Beeler, T., Tombari, F., Guibas, L. and Wetzstein, G., 2024, September. Physavatar: Learning the physics of dressed 3d avatars from visual observations. In European Conference on Computer Vision (pp. 262-284). Cham: Springer Nature Switzerland.

- Some key descriptions of the proposed method are missing:
    - How discrete constitutive models of each objects are set? If the types are given, "Predicted Categorical Distribution" in Fig 1 is very misleading. And [1] already handled this case. If the types are predicted, the paper does not mention the procedure at all in method section.
    - How silhouettes are rendered? Are they rasterized directly from MPM point clouds?
    - How are "extracted surfaces" (Line 256) extracted in detail?

- No real-world examples are provided.

**Questions:**

Key questions that needs justifications or clarifications are mentioned above.

---

> ### Author Response · Authors · 2025-11-26
>
> > The contribution is limited. The following two works are not cited.
>
> We thank the reviewer for highlighting these related works and have cited both in the revision. Our contributions are orthogonal rather than overlapping with theirs.
>
> > The task of multi-object system identification from videos is not novel.
>
> **Problem setting.** CoupNeRF [1] focuses on multi-material reconstruction in a free-falling setting where objects have known or zero initial velocities. This setup limits motion diversity and contact dynamics, resulting in relatively mild deformations and weak interactions. Our setting differs significantly. We study general multi-object dynamics with unknown initial velocities and frequent, contact-rich interactions, which introduce challenging occlusions, deformations, and complex energy transfers.
>
> **Methodology.** CoupNeRF [1] relies on implicit NeRF representations optimized over time. In highly dynamic, contact-heavy scenes, this leads to temporally inconsistent geometry and texture artifacts, since the radiance field struggles to represent large non-rigid changes. Our approach, with dynamic Gaussian representation, better handles large deformations and dense contacts.
>
> **Quantitative and qualitative results.** We reproduced CoupNeRF [1] on our benchmark and observed that it underperforms our MOSIV approach, particularly during impact phases and long-horizon rollouts due to the above limitations.
>
>
> | **Metric \\ Method**     | **CoupNeRF\*** | **OmniphysGS** | **MOSIV** |
> |--------------------------|----------------|----------------|-----------|
> | **Runtime (s) ↓**        | 9591.09        | 5263.70        | 5021.46   |
> | **GPU Memory (GB) ↓**    | 31.14          | 61.08          | 29.79     |
>
>
> **Table R1. Runtime and GPU memory consumption comparison.**
>
>
> | **Method**     | **PSNR ↑** | **SSIM ↑** | **CD ↓** | **EMD ↓** |
> |----------------|------------|------------|----------|-----------|
> | **OPGS**       | 25.93      | 0.945      | 11.79    | 0.095     |
> | **CoupNeRF\*** | 25.88      | 0.962      | 0.927    | 0.047     |
> | **MOSIV**      | 31.17      | 0.975      | 0.389    | 0.033     |
>
> **Table R2. Comparison of Observable state simulation performance on MOSIV dataset.**
>
> | **Method**     | **PSNR ↑** | **SSIM ↑** | **CD ↓** | **EMD ↓** |
> |----------------|------------|------------|----------|-----------|
> | **OPGS**       | 19.00      | 0.888      | 51.92    | 0.199     |
> | **CoupNeRF\*** | 20.62      | 0.942      | 1.859    | 0.062     |
> | **MOSIV**      | 28.92      | 0.964      | 0.894    | 0.050     |
>
> **Table R3. Comparison of Future state simulation performance on MOSIV dataset.**
>
> > Geometry-driven supervision is not novel.
>
> We acknowledge that geometry-driven supervision is widely adopted in 3D vision and is not our main contribution. Further, we highlight the key differences against PhysAvatar and key design choices in geometry-driven supervision we made specifically for our problem setting.
>
> **Task scope.** We address multi-object, contact-rich dynamics with unknown initial velocities and object interactions. PhysAvatar optimize a single system. We are solving a different task where object identities, contacts, and occlusions must be disambiguated throughout time.
>
> **Object-aware supervision.** We introduce per-object geometry and silhouette supervision that preserves object identities through contact. Losses are computed on disjoint supports with explicit cross-object exclusion, which prevents gradients from one object being explained away by a neighbor during collisions. This design is critical once contact happens and surfaces interpenetrate in image space.
>
> > How discrete constitutive models of each objects are set?
>
> We provide 2D material type segmentation masks and optimize per-Gaussian material type during dynamic reconstruction, similar to CoupNeRF [1]. "Predicted Categorical Distribution" in Fig. 1 refers to the optimized per-Gaussian material type distribution.
> We have updated Fig. 1 accordingly for clarity.
>
> > How silhouettes are rendered?
>
> Silhouettes are rendered from the simulated particles, which have been assigned Gaussian opacity during the conversion from Gaussians to occupancy volumes. In other words, particles carry both Gaussian attributes and physical properties, allowing it for both rendering and dynamic simulation.

---

> > ### Author Response · Authors · 2025-11-26
> >
> > > How are "extracted surfaces" (Line 256) extracted in detail?
> >
> > We follow GIC to obtain dense continuums as well as continuum surfaces. Specifically, the process operates in three stages.
> > 1. Internal volume initialization. We generate a rough internal shape by randomly sampling particles within the bounding box of Gaussian points and retaining only those that align with the object's depth as rendered from multiple camera views.
> > 2. Iterative field refinement. We construct a density field that progressively increases in resolution. In each iteration, we upsample the grid, smooths the field (mean filtering) to blur boundaries, and reassign high density to voxels containing actual particles to prevent the smoothing process from eroding the object's true shape.
> > 3. Surface extraction. Finally, the specific object surface is isolated by applying a threshold to this high-resolution, refined density field.
> > We have updated the accordingly for clarity.
> >
> > > No real-world examples are provided.
> >
> > We acknowledge the value of real-world evaluation. However, our primary contribution is the precise inverse identification of physical properties under complex contact conditions. Rigorously validating this requires absolute ground-truth constutive parameters (e.g., friction, elasticity) to distinguish between correct parameter recovery and mere visual overfitting. As measuring these internal properties is currently intractable in real-world scenarios, we prioritize a controlled synthetic benchmark, which is a common practice as in PAC-NeRF. With this theoretical foundation now established, we are excited to extend our framework to real-world applications as a key direction for future work.

---

> > > ### Author Response · Authors · 2025-11-28
> > > **Have we addressed all the issues?**
> > >
> > > Dear Reviewer,
> > >
> > > Thank you again for your thoughtful feedback. We have carefully addressed all the comments and questions raised in your earlier review. As the discussion phase is nearing its end, we would greatly appreciate it if you could let us know whether our responses satisfactorily resolve the issues you identified.
> > >
> > > We sincerely appreciate your consideration of a potential rating update once all concerns have been addressed.
> > >
> > > Thank you for your time and effort.

---

### Author Response · Authors · 2025-11-30
**Revision Summary**

We sincerely thank all reviewers for their insightful feedback. They find the vision appealing, dataset useful for community(yKeE), the method pioneering, effective, and robust (yotJ), and the results comprehensive and promising (dfDn).

To ensure the clarity of our method's presentation, we included discussion of CoupNeRF in Sec.1 and Sec.2, and We updated Figure 2 and the description of our two-stage pipeline in Sec.3. To address concerns about robustness and efficiency, we conducted extensive new experiments. We include Table 3 and Fig. 5 in Sec. 3 to provide a complete qualitative and quantitative comparison of our method against OmniPhysGS and CoupNeRF. In the Appendix, we add Table 6 comparing computation overhead. We also introduce the newly generated three-object dataset in Fig. 8 and report quantitative result of our method on this benchmark in Table 7.

---

### Meta-Review · Area_Chair_zkFh · 2026-01-07

**Summary:**

This paper received mixed reviews. The main concerns raised in the reviews are:
1. limited contribution and missing references (`dfDn`).
2. missing technical details and inadequate motivation for model design (`dfDn`, `yotJ`, `SAcW`).
3. limited evaluation, no real-world results (`dfDn`, `yKeE`, `SAcW`).

Overall, I quite like the motivation of this paper, but the current evaluations with simple synthetic scenes are very limited, as pointed out by several reviewers. That said, the experiments do show significant improvement over existing baselines. This is a borderline submission, and considering all factors, I think it is still acceptable but would strongly encourage the authors to expand the evaluations to make it stronger for future iterations.

**Reviewer Concerns:**

1. Concern #1 is partially addressed by additional discussion on the differences from the two existing works mentioned. Although the missing references do slightly undermine the claimed contributions, the arguments provided in the rebuttal seem quite reasonable. Additional quantitative comparisons also demonstrated significantly improved results.
2. I believe Concern #2 is also partially addressed by the detailed responses, although it is difficult to determine if all reviewers would be fully satisfied with the explanations.
3. Concern #3 is partially addressed by the new experiment results on three-object interactions. However, these results are still quite limited, and the rebuttal and the revision do not provide sufficient details of these additional experiments (e.g., how many samples, what types of objects, how are the interactions generated, etc.). More critically, the authors dismissed the request for real-world evaluation citing its difficulty. While I recognize the challenge, I still agree with the reviewers that the synthetic examples demonstrated in the paper are too simplistic. Demonstrating real-world results would greatly strengthen the contributions.

**Reviewer Scores:**

1. Reviewer `dfDn` (2->?): The reviewer might increase the rating, but it is unlikely that they would upgrade to a positive score.
2. Reviewer `yKeE` (6->6+): The reviewer might maintain or increase the rating.
3. Reviewer `yotJ` (6->6+): The reviewer might maintain or increase the rating.
4. Reviewer `SAcW` (6->6+): The reviewer might maintain or increase the rating.

---

### Decision · Program_Chairs · 2026-01-26

Accept (Poster)